# Spatiotemporal transcriptomic maps of whole mouse embryos at the onset of organogenesis

Abhishek Sampath Kumar [1,2,10], Luyi Tian[3,10], Adriano Bolondi [1,10], Amèlia Aragonés Hernández [1,4], Robert Stickels[3,5], Helene Kretzmer[1], Evan Murray[3], Lars Wittler[6], Maria Walther[1], Gabriel Barakat[1], Leah Haut[1,4], Yechiel Elkabetz [1], Evan Z. Macosko [3,7,11], Léo Guignard [8,11], Fei Chen [3,9,11] & Alexander Meissner [1,3,4,9,11] ✉

Spatiotemporal orchestration of gene expression is required for proper embryonic development. The use of single-cell technologies has begun to provide improved resolution of early regulatory dynamics, including detailed molecular definitions of most cell states during mouse embryogenesis. Here we used Slide-seq to build spatial transcriptomic maps of complete embryonic day (E) 8.5 and E9.0, and partial E9.5 embryos. To support their utility, we developed sc3D, a tool for reconstructing and exploring three-dimensional 'virtual embryos', which enables the quantitative investigation of regionalized gene expression patterns. Our measurements along the main embryonic axes of the developing neural tube revealed several previously unannotated genes with distinct spatial patterns. We also characterized the conflicting transcriptional identity of 'ectopic' neural tubes that emerge in *Tbx6* mutant embryos. Taken together, we present an experimental and computational framework for the spatiotemporal investigation of whole embryonic structures and mutant phenotypes.

Embryonic development necessitates the precise timing and location of numerous molecular, cellular and tissue-level processes[1–5]. These events are directed via spatiotemporal control of gene expression that orchestrates cell type specification, migration and localization[6–9]. Any disruption of this regulation often results in embryonic lethality or developmental defects[5,10,11]. At the end of gastrulation and the onset of organogenesis (embryonic days (E) 8.5–9.5), tissues experience major morphological changes, such as heart looping,

brain compartmentalization and neural tube folding, to guarantee proper structure and function[12–14]. Through neurulation, epithelial cells in the neural plate fold to form a morphologically defined tube, which exhibits a stratified gene expression signature along its dorsoventral (DV) axis, which is necessary for subsequent neuronal subtype diversification[15–21]. Many genes involved in this process have been identified, but the precise gene regulatory networks governing these patterns remain under investigation. Recent single-cell studies have begun to provide

[1]Department of Genome Regulation, Max Planck Institute for Molecular Genetics, Berlin, Germany. [2]Institute of Biotechnology, Technische Universität Berlin, Berlin, Germany. [3]Broad Institute of MIT and Harvard, Cambridge, MA, USA. [4]Institute of Chemistry and Biochemistry, Freie Universität Berlin, Berlin, Germany. [5]Graduate School of Arts and Sciences, Harvard University, Cambridge, MA, USA. [6]Department of Developmental Genetics, Max Planck Institute for Molecular Genetics, Berlin, Germany. [7]Department of Psychiatry, Massachusetts General Hospital, Boston, MA, USA. [8]Aix Marseille University, Toulon University, Centre National de la Recherche Scientifique, Laboratoire d'Informatique et Systèmes 7020, Turing Centre for Living Systems, Marseille, France. [9]Department of Stem Cell and Regenerative Biology, Harvard University, Cambridge, MA, USA. [10]These authors contributed equally: Abhishek Sampath Kumar, Luyi Tian, Adriano Bolondi. [11]These authors jointly supervised this work: Evan Z. Macosko, Léo Guignard, Fei Chen, Alexander Meissner. ✉e-mail: meissner@molgen.mpg.de

a deeper understanding of the topography of fate specification and highlighted some molecular mechanisms underlying these cell state transitions[22–28]. One limitation of dissociation-based approaches is their inability to preserve tissue structure, which precludes expression analysis within the native context. Recent advances in spatial transcriptomic technologies have begun to fill this gap, aiming to explore the organization of cell types within adult tissues and developing embryos[29–38].

In this study, we used Slide-seq, a technology that generates transcriptome-wide gene expression data at 10-µm spatial resolution[33,39], to build maps of whole embryos during early mouse organogenesis. Our data enabled the exploration of spatial gene expression patterns, cell state distributions, the reconstruction of three-dimensional (3D) transcriptomic maps for 'virtual' gene expression analysis and mutant phenotype dissection. We specifically leveraged the data to identify regionalized gene expression and differentiation trajectories in space, focusing on neural tube formation and patterning. Overall, we provide a comprehensive, high-resolution spatial atlas together with an accessible and ready-to-use visualizer to explore gene expression patterns in the developing mouse embryo.

## Results

### Spatial transcriptomic maps to construct 3D virtual embryos

To spatially map cell identities during early organogenesis, we used Slide-seq on two representative E8.5, one E9.0 and three partial E9.5 embryos (Fig. 1a and Extended Data Fig. 1a). For the two E8.5 embryos, we obtained 15 and 17 sagittal sections (10-µm thickness), respectively, with approximately 30-µm intervals between them. For the E9.0 embryo, 26 sagittal sections with 20-µm intervals, and for the three E9.5 embryos, 13 slices from the midline were obtained (Fig. 1a and Supplementary Table 1). In total, we recovered 533,116 high-quality beads with a median value of 1,798 transcripts and 1,224 genes per bead (Extended Data Fig. 1b–d). To ascertain the cell states assigned to each bead, we computationally mapped beads to a previously generated single-cell reference (Extended Data Fig. 1e,f)[26]. With this information, we extracted from each sequenced bead (1) spatial coordinates, (2) associated gene expression profile and (3) cell state assignment. Overall, we observed good alignment of cell states and spatial restriction of marker genes, such as *Ttn* (heart), *T* (tail bud), *Meox1* (somites) and *Sox2* (neural tube, brain), among others (Extended Data Fig. 2a–e). Additionally, we observed high reproducibility in recovering a comparable embryo composition and gene expression patterns among replicates (Extended Data Fig. 2c–f).

To translate our two-dimensional data into a 3D embryo, we developed sc3D, a computational method that enables the alignment of individual spatial transcriptomic arrays for 3D reconstruction. Specifically, we used sc3D to align serial Slide-seq 'z' samples from E8.5 and E9.0 embryos, which allowed us to capture the spatial distribution and morphologies of the emerging tissues at the onset of organogenesis (Extended Data Fig. 3a,b and Supplementary Figs. 1 and 2). This in turn enabled quantitative measurements of their volumes (250–39,264 × 10³ µm³), which were reproducible across individual replicate embryos (Extended Data Fig. 3c–e, Supplementary Fig. 1, Supplementary Movie 1 and Supplementary Table 2). We further showed that the reconstruction remained consistent as the interval between slices increased, with very minimal distortion in the rotation axes (Extended Data Fig. 3f). Importantly, sc3D also allows 3D reconstruction of other spatial transcriptomic datasets with high precision, speed and robustness to reduced spatial resolution[40] (Extended Data Fig. 4a–c and Supplementary Table 3).

We next used the reconstructed embryos to perform 'virtual' in situ hybridization (vISH) of over 27,000 genes on a quantitative scale, with the opportunity to query gradient gene expression along any given body axis, inclination plane and rotation angle (Fig. 1b–d, Supplementary Figs. 1 and 2, and Supplementary Movies 1–4). To further increase sc3D accessibility, we developed sc3D-viewer, a user-friendly

and interactive environment for exploring the Uniform Manifold Approximation and Projection (UMAP) representation, spatial cell state maps, and vISH for single and dual gene combinations (methods in Supplementary Information). To investigate spatial gene expression within each tissue type, we computed the genome-wide correlations between tissue volumes and densities of expressing cells to generate a localization score that allowed us to rank genes within each tissue based on their spatial restriction in expression. This analysis identified a set of highly informative, tissue-specific, regionalized genes within the embryo and across replicates as well as developmental stages (Figs. 1e,f and 2a, Extended Data Fig. 4d–g and Supplementary Table 4). For instance, we found high localization scores for genes such as *Nppa*, *Tdgf1*, *Cck* and *Sfrp5* in the developing heart tube (Extended Data Fig. 5a–d and Supplementary Movie 5). Mapping these genes onto our digital embryo revealed that their expression domains mark specific developmental axes (anterior–posterior (AP), DV and right-left) and delineate presumptive anatomical structures, such as the primitive ventricles and atria, the outflow tract, the cardiomyocyte jelly and the venous pole, respectively[41–43], as opposed to spatially ubiquitous gene expression seen in the blood (Extended Data Fig. 5c–e). Furthermore, *Cck* and *Sfrp5* expression spatially distinguish differentiated from undifferentiated cardiomyocytes domains (Extended Data Fig. 5d)[41–43]. Taken together, this demonstrates our ability to identify regionalized markers and study the distinct domain organization within complex developing tissues along developmental axes.

### Delineating molecular boundaries in the developing brain

Between E8.5 and E9.5, the most anterior portion of the neural tube develops into three distinct vesicles (prosencephalon–forebrain, mesencephalon–midbrain and rhombencephalon–hindbrain), which together form the primordial brain[20,21,44–47]. Our high-resolution spatial transcriptomic map of the forebrain–prosencephalon in E9.5 embryos allowed us to identify presumptive telencephalon and diencephalon regions and to delineate DV patterning of the diencephalon–midbrain (Extended Data Fig. 6a–e)[48]. To study the emergence of such patterns, we analyzed the transcriptome of E8.5, E9.0 and E9.5 brains and found that genes such as *Foxg1*, *Barhl2*, *Otx2*, *En1* and *Egr2* show regionalized expression patterns already at E8.5 (Figs. 1f and 2a). While *Foxg1* was confined to the rostral prosencephalon, defining the presumptive telencephalon, *Barhl2* was expressed caudally, already marking the presumptive diencephalon[49–51]. *Rax*, a marker of the developing eye, exhibited spatial gene expression confinement between E8.5 and E9.5, defining the future optic cup (Fig. 2a). It appears therefore that spatial restriction of gene expression precedes anatomical segregation. To further explore the relationship between cell fate commitment and spatial restriction of emerging structures, we used unsupervised spatial RNA velocity without prior knowledge of cell states[52]. We recovered distinct ranges of velocity dynamics with either converging or diverging trajectories, potentially corresponding to stepwise transitions or cellular steady states (Fig. 2b and Extended Data Fig. 7a,b). A closer inspection into low-velocity regions (defined by the low-velocity length and confidence of vector directionality), combined with the expression of known marker genes, revealed the presence of progenitor field domains (R1 anterior neural ridge) as well as differentiated neuronal territories (R5 hindbrain–spinal cord boundary) (Extended Data Figs. 6b,c and 7c). Additionally, areas with diverging trajectories highlighted boundary regions, such as the R4 mesencephalon–rhombencephalon boundary, which was marked by the restrictive and exclusive expression patterns of *Otx2* in the mesencephalon and *Gbx2* in the rhombencephalon, as well as *Fgf8* at the boundary (Extended Data Fig. 7c)[53].

Although vector ends do not necessarily represent a terminally differentiated state, such a relationship might be inferred when spatial trajectories are known to match with developmental patterning processes. For instance, the trajectories observed at R4 resemble the lineage specification of the mid–hindbrain progenitors that

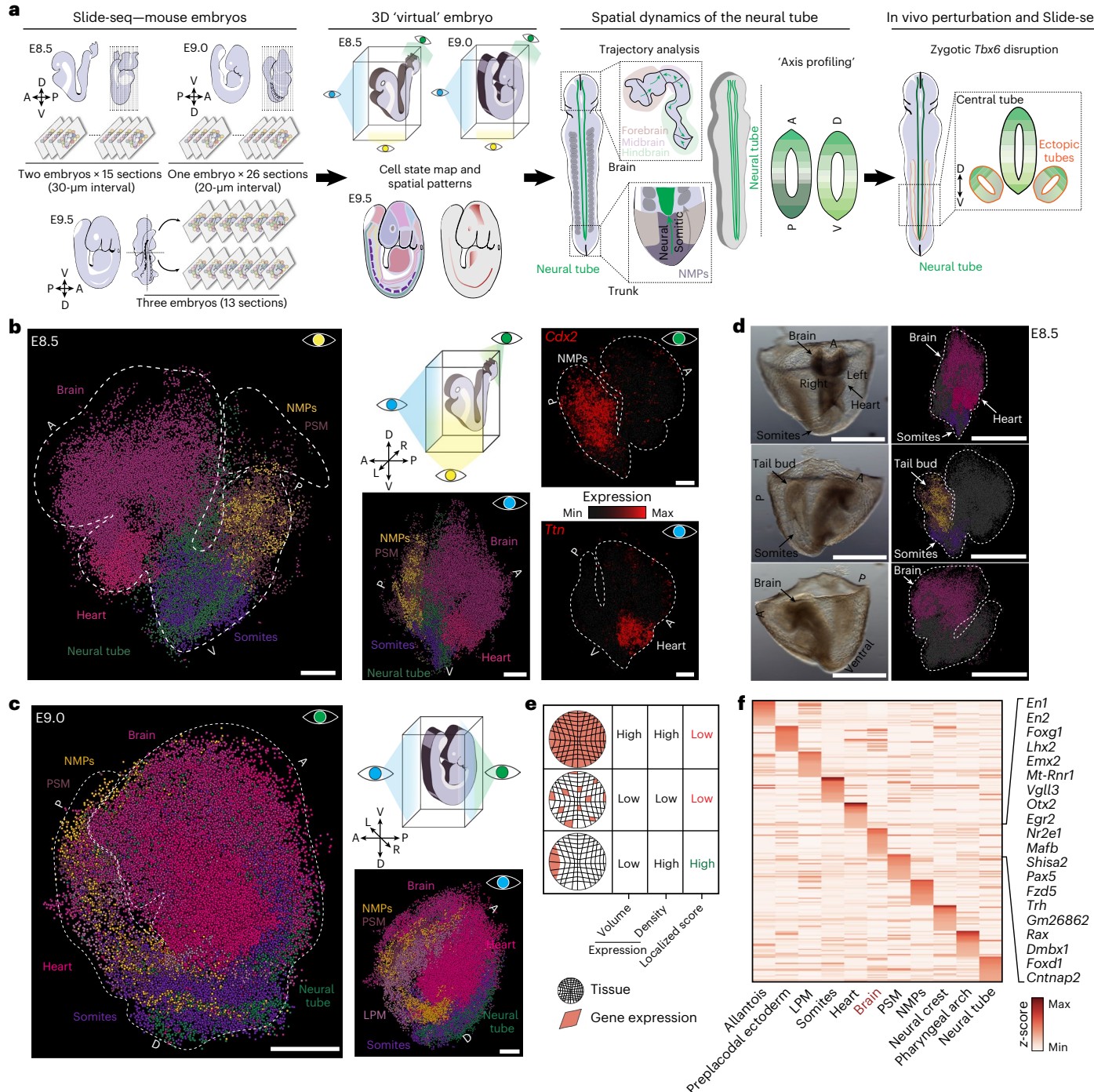

**Fig. 1 | Embryo-wide profiling of gene expression with spatial coordinates using Slide-seq. a**, Schematic of the experimental workflow and data analysis. Sagittal sections of mouse embryos at E8.5, E9.0 and E9.5 were obtained for Slide-seq. The dotted lines indicate the approximate position of the embryonic sections. **b,c**, 3D-reconstructed E8.5 (**b**) and E9.0 (**c**) stage embryos with six cell states highlighted (brain, heart, neural tube, somites, NMPs, PSM); the caudal marker gene *Cdx2* and heart tube marker gene *Ttn* are shown (normalized gene expression) in a vISH of the reconstructed E8.5 embryo1. Each dot corresponds to a single bead. Point of view is denoted by the eye symbol. Scale bars, 100 μm.

**d**, 3D view of an E8.5 embryo showing the indicated cell states and anatomical features in brightfield (left) or mapped onto the E8.5 embryo1 (right). Different orientations are displayed. Scale bars, 500 μm. **e**, Schematic of the strategy used to identify localized gene expression in tissues. **f**, Heatmap of localization scores for the top 20 spatially differentially expressed genes in 3D for each analyzed tissue in the E8.5 embryo1 (see Supplementary Table 4 for the list). Top 20 genes (row *z*-score-normalized) in the brain are highlighted. A, anterior; LPM, lateral plate mesoderm; P, posterior; D, dorsal; V, ventral; LPM, lateral plate mesoderm; NMPs, neuromesodermal progenitors; PSM, pre-somitic mesoderm.

will generate neuronal cells that subsequently populate the whole midbrain and anterior hindbrain areas[21]. In addition, our high-resolution map enabled a more granular view of the molecular determinants of developing boundaries before the formation of anatomical constriction (Fig. 2c and Supplementary Table 5). We analyzed the three boundaries demarcating the main brain regions to identify features that might shed light on the regulatory mechanisms involved in brain regionalization. In R2 and R4, we identified several signaling

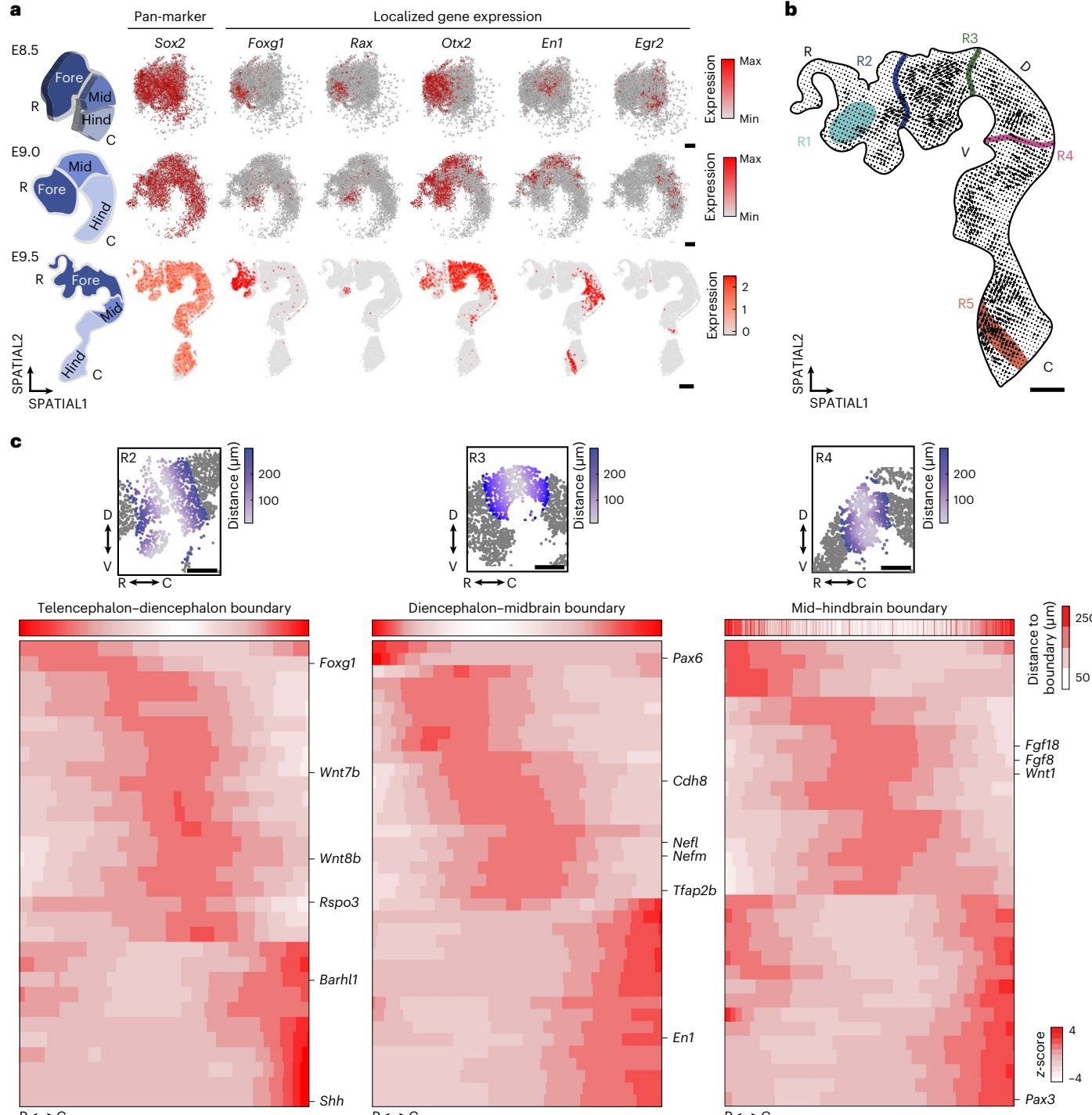

**Fig. 2 | Localized gene expression patterns in the developing brain.**
**a**, Schematic (left) and normalized gene expression spatial plot (right) of selected genes (highlighted in Fig. 1f) in the 3D brain of E8.5 embryo1, E9.0 and E9.5 (array E9.5_3). Each dot represents a single bead. **b**, Spatial plot of RNA velocity in the E9.5 (array E9.5_3) brain region. Vector direction indicates the trajectory and length denotes the magnitude. Low-velocity regions are indicated as R1, R2, R3 and R4. R1, neural ridge prosencephalon; R2, telencephalon–diencephalon boundary; R3, diencephalon–mesencephalon boundary; R4: mesencephalon–hindbrain boundary; and R5, hindbrain–spinal cord boundary. **c**, Spatial plot showing the brain boundaries (top) and top enriched spatially differentially expressed genes along the R2, R3 and R4 brain boundaries (represented as row z-score-normalized expression) at E9.5 (array E9.5_3). The full list of genes can be found in Supplementary Table 5. Scale bar for all plots, 50 µm. C, caudal; D, dorsal; R, rostral; V, ventral.

molecules (WNT, FGF) and downstream effectors ratifying the role of these boundaries as signaling centers instructing the patterning of the adjacent structures (the zona limitans intrathalamica (ZLI) and the isthmic organizer). While the interplay between WNT and FGF signaling in the mid–hindbrain boundary (R4) is well known, their role at the telencephalon–diencephalon boundary (R2), where SHH signaling has a major role[54], has been less studied. In this study, we show the spatial constriction of *Wnt7b* expression to the boundary together with *Wnt8b* (Extended Data Fig. 7d). Although *Wnt7b* is expressed in the rostral and dorsal parts of the diencephalon[55], its co-expression and

colocalization with *Wnt8b*, known to be expressed within the ZLI[56], has not been described yet. Compared to the other analyzed regions, the diencephalon–midbrain boundary (R3) is characterized by low signaling molecule activity, high neuronal marker expression (neurofilament proteins and neuronal genes) and convergent RNA velocity signature, suggesting the presence of a more mature neuronal rather than a progenitor domain (Fig. 2c and Supplementary Table 5). Therefore, our analysis can help identify relevant molecules such as CDH8 (Fig. 2c and Supplementary Table 5), which is known to compartmentalize the diencephalon–midbrain boundary together with other cadherins[57]. Moreover, within the forebrain region, we mapped known as well as uncharacterized gene expression distributions, including those involved in eye development (Extended Data Fig. 7e,f). By examining such patterns during early brain development, we stratified the spatial emergence of anatomical structures.

## Cell identity and spatial distance in the trunk

While the anterior neural tube develops into the brain, the posterior portion generates the future spinal cord during trunk elongation. The embryonic trunk consists of morphologically diverse structures with distinct developmental origins that ensure correct axial elongation and body plan segmentation (Fig. 3a)[58–60]. As the embryo develops, axial progenitor cells acquire a bipotent differentiation potential that allows the generation of neuroectodermal and mesodermal derivatives[58–60]. Specifically, these cells, known as neuromesodermal progenitors (NMPs), generate the posterior portion of the neural tube and the paraxial mesoderm via the determination front (pre-somitic (PSM) and somitic mesoderm) (Fig. 3a)[58–60]. To profile gene expression regionalization in the developing trunk in 3D, we first mapped the cell states corresponding to the NMPs, PSM and somites (somitogenesis trajectory) onto our E8.5 virtual embryo (Fig. 3b). The vISH of *Tbx6*, *Ripply2* and *Meox1* further confirmed the organized gene expression patterns along the AP and right-left symmetry involved in somitogenesis (Fig. 3b). Next, to understand how a regionalized gene expression signature impacts developmental dynamics, we profiled trunk developmental trajectories by combining transcriptional pseudotime measurements with spatial information. UMAP and spatial visualization revealed a continuum of transcriptomic states along the somitic and neural trajectories (Fig. 3c,d and Extended Data Fig. 8a–c). We next investigated this relationship in more detail and calculated the transcriptional and spatial distances for every bead to every other bead within each trunk tissue in an E9.5 embryo ('pseudotime' and 'spatial' distances, respectively) (Fig. 3e)[61–63]. Interestingly, we discovered that changes in transcriptomes are not necessarily proportional to cell–cell distances and that their relationship is tissue-specific. In particular, progenitor cells (NMPs and PSM) occupy a small region of the embryo, showing low spatial distance distributions concurrent with low transcriptional variability (Fig. 3e). On the other hand, more differentiated cells (neural tube and somites) are widely distributed along the trunk region (high distance distribution), even in cases of low transcriptional differences (Fig. 3e). We also observed a group of proximal cells in the neural tube characterized by notable transcriptional variability that, when mapped to spatial arrays, represented local DV patterning (Fig. 3e dotted line and Fig. 3f). Together, our findings show that during trunk development, progenitors differentiate into subsequent cell states in a restricted spatial domain before dispersal.

## Emerging patterns along the neural tube axes

As the trunk develops and tissues extend in space, the transcriptional differences along their length determine the positional identity of various cellular states. For example, between E8.5 and E10.5, the neural tube folds from the neural plate and undergoes patterning to establish the cellular stratification for the future spinal cord[16,17,19–21,62–64]. Regionalized gene expression programs guarantee further diversification of neuronal types along the AP and DV axes[16,17,19–21,62–64]. To understand the genetic programs involved in establishing the discrete progenitor

domains along the neural tube, we isolated the corresponding beads and searched for spatial co-expression patterns along the AP (approximately 4,600 μm) and DV (approximately 320 μm) axes ('axis profiling') (Fig. 4a and Extended Data Fig. 8d). We identified distinct expression patterns along the neural tube AP axis for genes involved in several cellular and molecular functions (Extended Data Fig. 8e). As expected, *Hox* genes were among the most highly localized genes within this axis (Fig. 4b)[65,66]. HOX factors interact with each other to regulate transcriptional programs. To identify putative functionally collinear groups, we performed *Hox* gene module analysis combined with spatial resolution and found six distinct modules of *Hox* gene expression, from the most anterior module (*Hox* module I, comprising *Hoxb2* and *Hoxa3*) to the most posterior one (*Hox* module VI, containing *Hoxd8* and *Hoxa9* (Fig. 4c and Extended Data Fig. 8f,g). Next, we examined the DV axis of the neural tube and were able to resolve and annotate well-studied structures like the notochord, the floor plate, the actual neural tube and the roof plate based on their spatially restricted transcriptional signature (Fig. 4a,d and Extended Data Fig. 9a,b)[29]. Among the most spatially constrained genes, we found well-known lineage-defining markers like *Zic1*, *Pax3*, *Olig2*, *Nkx6-1* and *Nkx2-9*, which we further confirmed using RNA–fluorescence in situ hybridization (FISH) (Fig. 4d–f and Extended Data Fig. 9c–e). We also identified 43 additional genes in the early mouse neural tube at the E9.5 stage that appear to exhibit a patterned expression along the DV axis in the ventricular zone containing progenitors (Fig. 4d–f, Extended Data Fig. 9f and Supplementary Table 7). Our results show the utility of the axis profiling tool in detecting well-studied gene expression patterns along the neural tube axis and its application in the de novo discovery of genes with locally restricted expression.

## Conflicting identity of ectopic neural tubes in *Tbx6* mutants

After cataloging the spatial transcriptome underlying the developing neural tube axes, we investigated a classic embryonic mutant phenotype where ectopic neural tubes arise. The T-box transcription factor TBX6 is expressed in the PSM and is required for somite segmentation and specification[10]. In embryos lacking *Tbx6*, ectopic neural tubes arise at the expense of the somitic compartments (Fig. 5a)[11,67]. To assess the precise molecular identity of the ectopic neural tubes, we used CAS9-based disruption of *Tbx6* in zygotes and performed Slide-seq on E9.5 wild-type (WT) and *Tbx6* mutant (*Tbx6* knockout (KO)) transversal embryo sections, focusing on the posterior segment of the trunk region where multiple tubes arise in the absence of *Tbx6* (Fig. 5a and Extended Data Fig. 10a–c)[26,68–70]. As expected, we observed an overrepresentation of beads assigned to the neural tube cluster in *Tbx6* KO embryos compared to WT controls, along with a commensurate lack of somitic cells (Fig. 5a).

Next, we reclustered the beads having a neural and somitic identity to more closely inspect differences between the neural tubes in the *Tbx6* KO embryos. We found that neural tube cells are subsequently resolved into four transcriptional subclusters, which we labeled as neural crest, neural plate and two main neural tube clusters (neural tubes 1 and 2, Extended Data Fig. 10d)[71–75]. Interestingly, when we spatially assigned the two neural tube cluster cells on WT and *Tbx6* KO arrays, we discovered that neural tube 1 cells mapped to the central tube in both genotypes. In contrast, neural tube 2 cells mapped to both sides of the central tube exclusively in the *Tbx6* KO, suggesting that ectopic tubes are characterized by a distinct transcriptional state (Fig. 5b). Specifically, cells of ectopic tubes display a transcriptional identity that is in between the somitic and neural cells, despite their predicted neural identity (Fig. 5c and Extended Data Fig. 10e,f). We observed high levels of mesodermal-specific genes including *Aldh1a2* and *Mest* (Fig. 5d,e). Concomitant with the expression of mesodermal markers, ectopic tube-assigned cells had reduced or absent expression of classic neural tube patterning marker genes such as *Olig2*, *Gm38103*, *Sox3*, *Nkx6-1* and *Shh* (Fig. 5d,e). Nonetheless, the mesodermal signature was

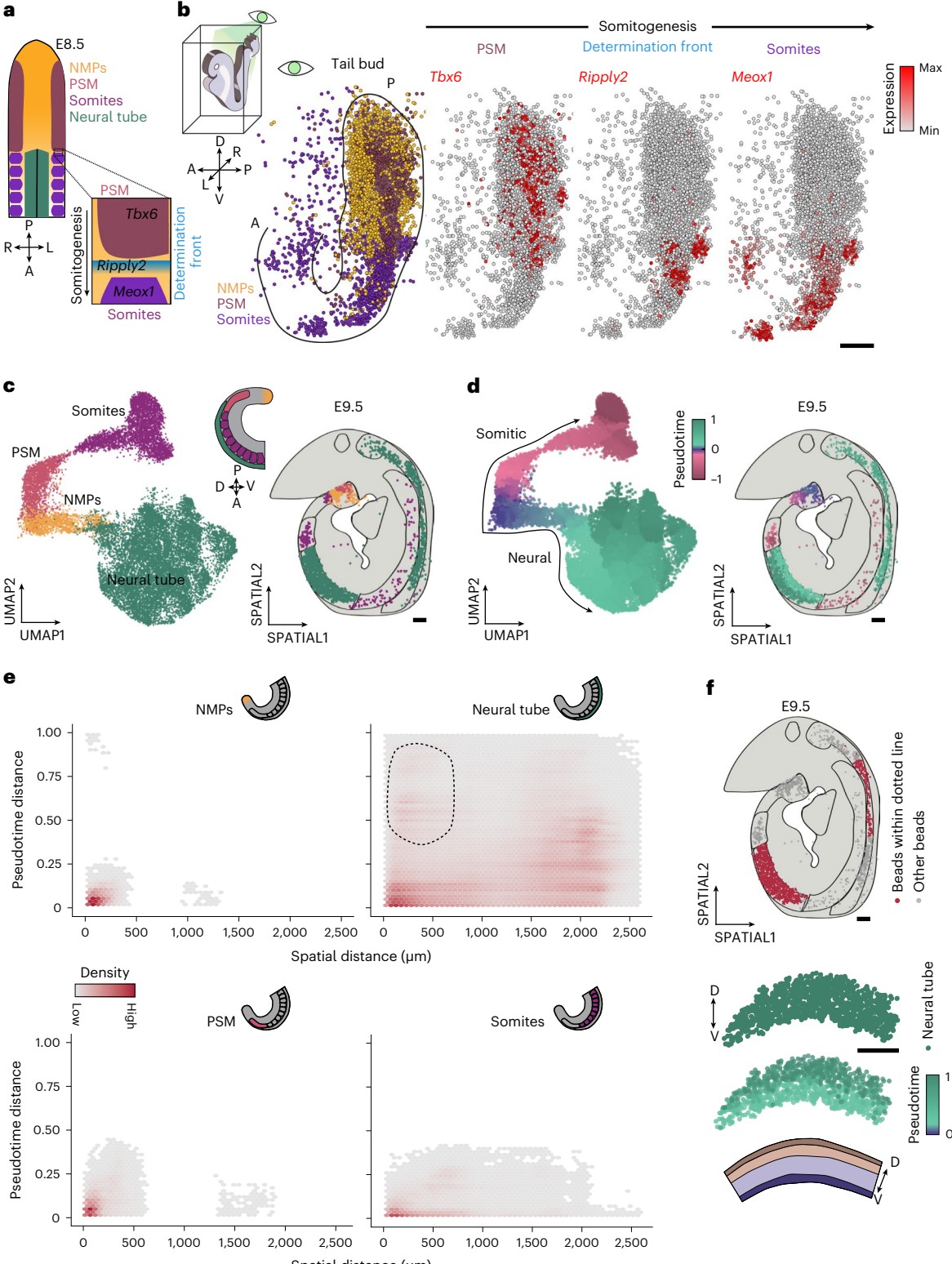

**Fig. 3 | Spatial organization of the embryonic trunk. a**, Schematic of the spatial organization of cell states in the embryonic trunk region at E8.5, and close-up view on the somitogenesis process. **b**, Schematic and 3D spatial plot of E8.5 (embryo2), showing selected cell states and vISH of *Tbx6*, *Ripply2* and *Meox1* in the trunk region. Each dot denotes a bead and the color corresponds to the indicated state. Scale bar, 200 μm. **c**, UMAP showing beads from E8.5 and E9.5 embryos corresponding to the NMPs, PSM, somites and neural tube states (left) and the corresponding spatial distribution (right) in an E9.5 embryo (array E9.5_2). Each dot denotes a bead and the color corresponds to the indicated state. Scale bar, 100 μm. **d**, UMAP showing pseudotime analysis along the somitic

and neural differentiation trajectories (left) and the corresponding spatial distribution (right) in an E9.5 embryo (array E9.5_2). Each dot denotes a bead and the color corresponds to the assigned pseudotime value. Scale bar, 100 μm. **e**, Density plot displaying the computed pseudotime difference (*y* axis) versus the measured spatial distance (*x* axis) between all beads of the same cell state. Each dot is a pairwise comparison. The dotted line delineates cells with low spatial distance and large transcriptional divergence in the neural tube. **f**, Spatial plot showing the beads (top) within the dotted line (Fig. 2e) and the distribution of pseudotime values in an E9.5 embryonic trunk (array E9.5_3) that reflects neural tube patterning. Each dot represents a bead. Scale bars, 100 μm.

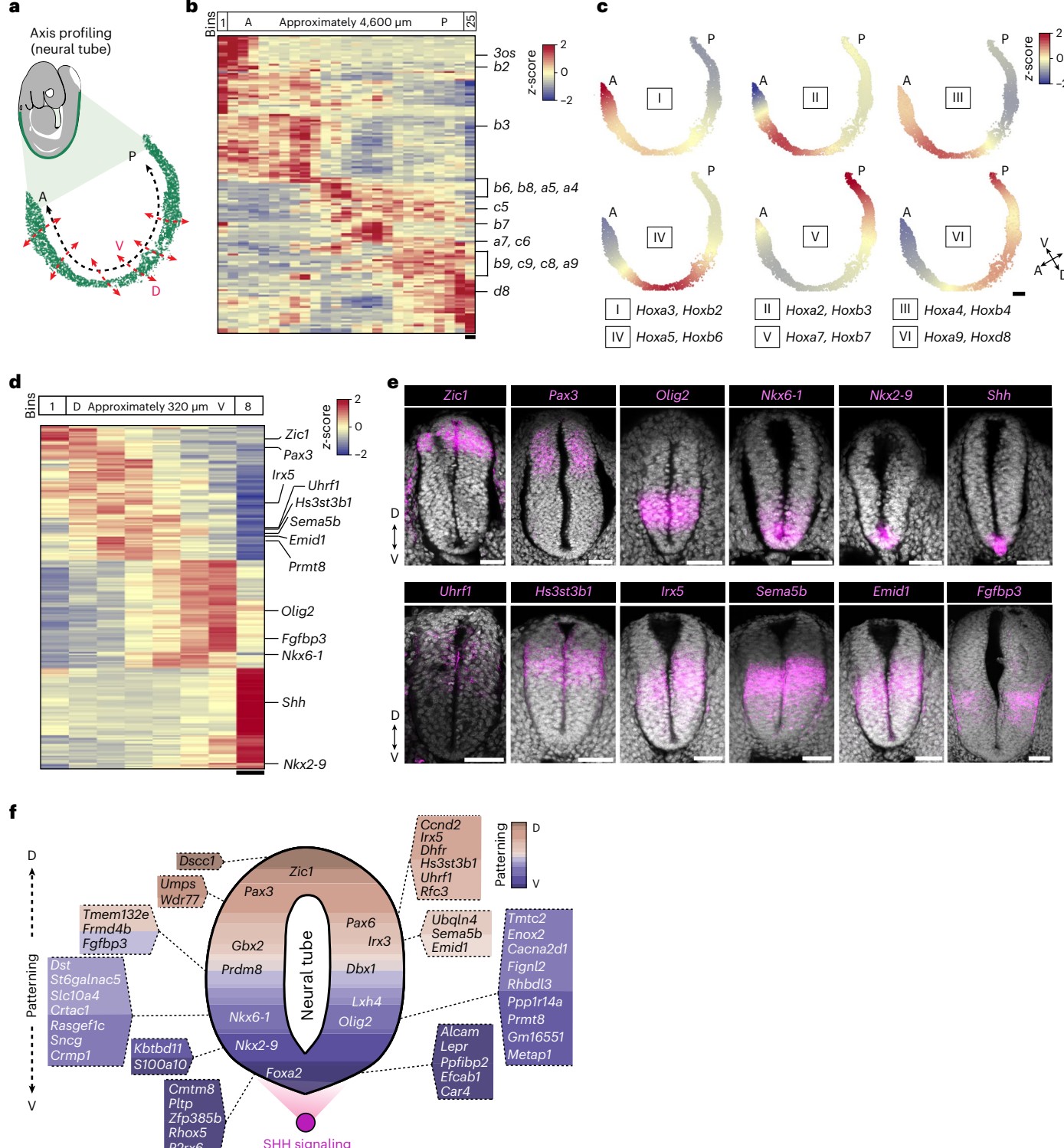

**Fig. 4 | Neural tube profiling along the AP and DV axes. a**, Schematic of an E9.5 stage embryo, with beads corresponding to the neural tube (green) and the denoted axes along which profiling was performed (array E9.5_10). **b**, Heatmap showing the top 160 genes with expression patterns (row z-score-normalized, filtered for a log fold change greater than 0.05 and a false discovery rate (FDR) < 0.01; see Supplementary Table 7 for the list of genes) in the 25 generated spatial bins along the AP axis. Expression of *Hox* genes is highlighted (right). Scale bar, 184 μm. **c**, Spatial plots showing the normalized expression (bin z-score) of *Hox* modules (I, II, III, IV, V, VI) in the E9.5 (array E9.5_10) neural tube. Two representative genes for each module are indicated. Scale bar, 100 μm. **d**, Heatmap showing the top 160 genes with expression

patterns (row z-score-normalized, filtered for a log fold change greater than 0.05 and FDR < 0.01; see Supplementary Table 7 for the list of genes) in the eight generated spatial bins along the DV axis. Expression of known patterned and our identified genes are highlighted. Scale bar, 40 μm. **e**, RNA–FISH showing validation for known (top) and our identified (bottom) patterns along the DV axis of the neural tube. A single tissue slice obtained from the posterior portion of the neural tube is shown in the images. *n* = 3 embryos with reproducible staining pattern for each profiled target from WT embryos. Scale bars, 50 μm. **f**, Schematic showing the spatial patterns of known (within the tube) and our identified (outside the tube) genes along the DV axis of the E9.5 stage neural tube.

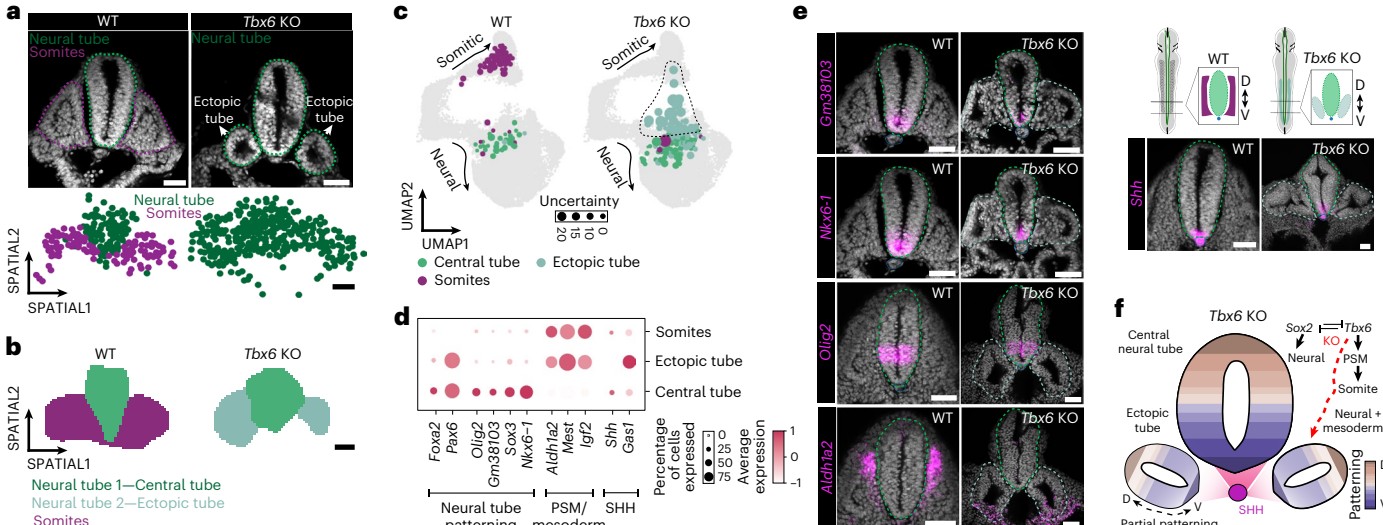

**Fig. 5 | Slide-seq profiling of *Tbx6* KO embryos. a**, Transverse section of DAPI stained embryos (top) and spatial plot (bottom) showing the annotated tissue morphologies (somites, neural tube) and corresponding cell states in E9.5 WT and KO embryo. The KO experiment was independently performed five times in total with *n* > 5 embryos per experiment, and consistently yielded the same phenotype. The Slide-seq experiment was performed on one representative transversal section for WT and two for KO embryos. **b**, Spatial grid map showing the organization of neural tube 1, 2 and somitic cells in WT and *Tbx6* KO embryos. **c**, UMAP showing the projection of cells assigned to the indicated clusters on the trunk trajectory (Fig. 3c), with the size of the dots representing the degree of uncertainty of mapping to the respective position. **d**, Dot plot showing the expression of the indicated genes in the three clusters (dot size is percentage of cells per cluster; color is cluster average normalized expression). **e**, RNA–FISH of the indicated genes in a transversal section of an E9.5 WT and KO embryos. The dotted lines in the schematic denote the AP position within the trunk from which sections were obtained. A representative section from WT and KO embryos at E9.5 is shown. The expression pattern was verified in two of the KO experiments with *n* > 3 embryos per experiment. Scale bars, 50 μm. **f**, Schematic showing the transcriptional identity and patterning characteristic of the ectopic neural tubes that arise in the absence of *Tbx6* expression, highlighting their conflicting transcriptional identity.

overlayed with the conflicting expression of other characteristic neural tube patterning genes, such as *Foxa2* and *Pax6* (Extended Data Fig. 10g)[11].

In summary, our spatial transcriptomic analysis of the ectopic tubes in *Tbx6* mutant embryos identified cells that acquire a mixed transcriptomic identity, characterized by the expression of mesodermal genes and partial DV gene expression patterning (Fig. 5f).

## Discussion

We performed embryo-wide spatial transcriptomic profiling using Slide-seq to decipher the tightly regulated gene expression patterns of approximately 27,000 genes in developing embryos at the onset of organogenesis. The reconstruction of digital 3D embryos using sc3D enabled the quantitative exploration of gene expression patterns and gradients on a virtual in situ basis. Combined with the development of sc3D-viewer (a napari plugin)[76], an accessible, interactive and user-friendly visualization platform to register and explore 3D spatial genomic data, we facilitate the rapid and seamless exploration of cell type distribution and gene expression patterns along any given developmental axis, including the possibility to reconstruct tissues from other spatial transcriptomic datasets.

Spatially resolved single-cell sequencing methods continue to evolve rapidly[37,38,40,77–79]. The increase of available datasets will require faster and more precise computational approaches to take full advantage of the added spatial information. sc3D contributes toward the in-depth analysis of the topology and geometry of gene expression and co-expression patterns, providing the infrastructure to start modeling gene expression diffusion, and their interaction in embryonic tissues. The sc3D data structure has been purposely designed to be close to the one of imaging-based cell tracking algorithm outputs. This similarity will ease the porting of cell tracking-based inter-sample alignment algorithms, such as Tardis, to the spatial transcriptomic field[31].

The dynamics at which transcriptomes evolve as progenitors differentiate in their spatial distribution has been challenging to explore. The analysis of differentiation trajectories in the embryonic brain and trunk regions revealed several discrete domains in which transcriptional changes converge or diverge spatially, indicating a non-linear and tissue-specific relationship. We observed that spatially dependent relationships were frequently associated with regions characterized by high intercellular signaling gradients. These changes may reflect the specification of sublineages, progenitor pool migration, differentiated subtypes maturation or regionalization of specific fates. Additional studies combining lineage recording, spatial information and single-cell transcriptomic profiling might help resolve the causal relationship between gene expression programs, spatial allocation and cell fate specification.

The transcriptional rewiring along the AP and DV axes of the developing neural tube defines discrete gene expression domains instrumental in controlling future cellular diversification[16,17,19–21,62–64]. We discovered several interesting genes that display regionalized gene expression patterns along these axes, including epigenetic and metabolic regulators that require further investigation to determine their molecular role and functional implications.

As an example for leveraging spatial information in a perturbation experiment, we provided a detailed transcriptomic characterization of the molecular identity of the ectopic neural tubes that arise in *Tbx6* mutant embryos. Unexpectedly, what has been historically assigned as additional neural tubes are morphologically tubular structures with incomplete patterning and continued expression of mesodermal genes that are usually associated with non-epithelial and mesenchymal cell identities. This suggests a decoupling of transcriptional programs and morphogenetic outcomes during embryonic development, with signaling gradients, extracellular matrix and mechanical clues probably playing crucial roles[80]. While *Tbx6* mutant embryos exhibit a distinct

and well-characterized phenotype, many other genes may cause less obvious morphological changes and hence will benefit from a similar spatiotemporal characterization to define their developmental roles.

Our accessible resource of spatial transcriptomic maps at the onset of organogenesis and supporting computational tools will help the continued exploration of mammalian development. Furthermore, the framework presented in this study could be implemented to conduct molecular spatial phenotyping on many additional perturbations. Lastly, by combining lineage mapping and multi-omic analysis, a comprehensive map of the gene regulatory network during embryogenesis could be developed based on this work.

## Online content

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

## Methods

All experiments described in this article comply with the relevant ethical regulations at the respective institutions. All experiments were approved by the Landesamt für Gesundheit und Soziales. All animal procedures were performed according to animal welfare guidelines and regulations approved by the Max Planck Institute for Molecular Genetics (G0243/18-SGr1_G and ZH120).

### Animal work and embryo preparation (WT and KO embryos)

WT E8.5, E9.0 and E9.5 embryos were dissected from the uteri of naturally mated CD-1 mice in 1× HBSS (catalog no. 14175053, Gibco) on ice. Embryos were staged based on morphology, size and somite number (3–5 somite pair stage for E8.5, 10–12 somite pair stage for E9.0 and 15–18 somite pair stage for E9.5). Extra-embryonic tissues were removed from E9.5 embryos before processing. Embryos were washed in cold 1× HBSS with 2 U ml$^{-1}$ RNase inhibitor (catalog no. N8080119, Thermo Fisher Scientific) and embedded in O.C.T. solution (catalog nos. 23-730-571 and 23-730-572, Thermo Fisher Scientific). Embryos in O.C.T. were oriented under a stereoscope, immediately placed on dry ice for flash-freezing and ultimately stored at −80 °C. *Tbx6* mutant embryos were generated by a previously established protocol[26,68,69,70]. Briefly, in vitro fertilized (IVF) zygotes were electroporated with Alt-R CRISPR–Cas9 RNP complex with guides targeting three different exons of *Tbx6* (Supplementary Table 10). Embryos that developed to blastocyst stage were retransferred to CD-1 pseudo-pregnant surrogate animals as described. All *Tbx6* mutant embryos isolated at E9.5 showed the mutant phenotype (enlarged tail bud, ectopic neural tubes). The trunk region was dissected from WT and mutant embryos (removed parts above the limb bud and heart), embedded in O.C.T. solution and frozen at −80 °C.

Animals were kept under specific pathogen-free conditions in individually ventilated cages at 22 ± 2 °C, 55 ± 10% humidity with a 12-h light–dark cycle (6:00–18:00). IVF was performed with B6D2F1 oocyte donors (aged 7–9 weeks; Envigo) and sperm was isolated from B6.CAST F1 males (aged 2 months, generated in-house by breeding C57BL/6J females and CAST/EiJ males). For the embryo transfer experiments, pseudopregnant CD-1 female mice (Hsd:ICR; 9–12 weeks old; 21–25 g; Envigo) were mated with vasectomized males (Swiss Webster; older than 13 weeks; Envigo).

### Cryosectioning for Slide-seq V2

Fresh-frozen O.C.T. blocks with mouse embryos were equilibrated to −20 °C in a cryostat (CM1950, Leica Biosystems), mounted onto a cutting block with O.C.T., sliced at a 10-μm thickness and then overlaid and melted onto sequenced spatial arrays[33,39]. Sagittal sections for whole embryos were collected at the following intervals: E8.5 embryos, 30-μm distance; E9.0 embryos, 20-μm distance; and for E9.5, sections from mid-volume were collected from three independent embryos. One for the head region and another for the thoracic and trunk region from each embryo (Fig. 1a and Supplementary Table 1). An E9.5 transversal section of the trunk region (posterior trunk corresponding to the somite–neural tube region) was collected for WT and *Tbx6* mutant embryo with a 10-μm section thickness.

### Whole-embryo RNA–FISH

Whole-embryo RNA–FISH was performed according to the protocol from Molecular Instruments with some modifications. Briefly, embryos fixed overnight with 4% paraformaldehyde (PFA) at 4 °C were washed three times for 10 min each with 1× PBS with 0.1% Tween 20 (PBST) at 4 °C. Embryos were dehydrated in an increasing concentration series of methanol + PBST washes, for 10 min each wash at 4 °C (25% methanol; 50% methanol; 75% methanol; 100% methanol). Embryos were stored at −20 °C overnight or longer. Next, embryos were rehydrated in a decreasing concentration series of methanol + PBST washes, for 10 min each wash at 4 °C (100% methanol; 75% methanol; 50% methanol; 25%

methanol; 100% PBST). After two washes for 10 min at 4 °C in PBST, embryos were bleached with 6% hydrogen peroxide (for endogenous peroxidase activity in blood cells) at 4 °C for 20 min. After two washes in PBST for 10 min each at 4 °C, embryos were treated with 10 μg ml$^{-1}$ proteinase K (catalog no. EO0491, Thermo Fisher Scientific) for the indicated time at room temperature (E9.5: 10 min). After two washes with PBST for 15 min each, embryos were postfixed in 4% PFA for 15 min at room temperature and washed three times in PBST for 15 min each step. Embryos were then prepared for hybridization by incubating in hybridization buffer at 37 °C for 1 h. Probes were resuspended in hybridization buffer at a concentration of 1 pM and incubated with embryos overnight at 37 °C. Embryos were washed four times with probe wash buffer for 15 min each wash at 37 °C, followed by three washes in 5× SSCT hybridization buffer + 0.1% Tween 20. Fluorescent hairpins were prepared as described by the manufacturer at a concentration of 0.06 μM each hairpin in amplification buffer. Embryos were then incubated in amplification buffer before incubation with hairpin probes overnight at room temperature in the dark. Excess probes were removed by five washes of 15 min each step in 5× SSCT at room temperature in the dark. Nuclei were counterstained by incubation with 2 μg ml$^{-1}$ 4,6-diamidino-2-phenylindole. The buffers and probe sequences used in this study are available at Molecular Instruments and their unique ID can be found in Supplementary Table 10. After hybridization, embryos were embedded in O.C.T. and frozen at −80 °C. O.C.T. blocks were sectioned to obtain transversal sections of the trunk region and neural tube at 30-μm thickness. Embryos and sections were imaged with a ZEISS LSM-880 confocal microscope at 10×, 20× magnification, averaging four times per frame and 10-μm z-stacks. Images were processed with the ImageJ software. The Plot Profile function was used to perform the signal intensity along a user-defined axis for each fluorescent channel.

### Slide-seq V2

The Slide-seq V2 protocol was used to generate all the sequencing libraries. Bead synthesis, array sequencing, image processing and base calling were performed[33,39] as described below. Briefly, the 10-μm barcoded beads were synthesized in-house by ChemGenes with a 14-bp spatial barcode separated by a 14-bp linker sequence, an 8-bp unique molecular identifier (UMI) sequence and a 20-bp poly(T) tail. The bead arrays were prepared by resuspending the synthesized beads in 10% dimethylsulfoxide at a concentration of 20,000–50,000 beads per microliter. Then, 10 μl of the bead solution was pipetted into each position on the gasket. The coverslip-gasket filled with bead solution was centrifuged at 750*g* for at least 30 min at 40 °C until the surface was dry. To extract the spatial barcodes, arrays were sequenced using Bioptechs FCS2 flow cells with an RP-1 peristaltic pump (Rainin) and a modular valve positioner (Hamilton MVP). During sequencing, flow rates between 1 and 3 ml min$^{-1}$ were used. Imaging was obtained with Nikon Plan Apo 10×/0.45 objective. Sequencing was performed using a sequencing-by-ligation approach. Base calling from the images was performed using the custom MATLAB package PuckCaller (https://github.com/MacoskoLab/PuckCaller).

### Slide-seq V2 library generation

The complete protocol for the Slide-seq V2 library preparation can be found at https://www.protocols.io/view/library-generation-using-slide-seqv2-81wgb7631vpk/v1?version_warning=no. Briefly, arrays covered with freshly cut tissue sections were transferred to tubes containing 6× SSC supplemented with RNAase inhibitor (1:20 concentration, catalog no. 30281-2, NxGen, Lucigen) and incubated for 15 min at room temperature. Arrays were then dipped in 1× reverse transcriptase buffer and then transferred to tubes containing the reverse transcription mix (Maxima 1× reverse transcriptase buffer, 1 mM deoxynucleoside triphosphates (dNTPs), 2 U ml$^{-1}$ RNase inhibitor, 2.5 mM template switch oligonucleotides (catalog no. 339414YCO0076714,

QIAGEN) and 10 U ml$^{-1}$ Maxima H minus reverse transcriptase) for 30 min at room temperature followed by a 90-min incubation at 52 °C. Proteinase K (1:50 concentration) and tissue clearing solution were added to the same tube and incubated at 37 °C for 30 min. Beads were then removed from the glass slide by pipetting up and down a few times and resuspended in TE-TW solution (10 mM Tris, pH 8.0, 1 mM EDTA, 0.01% Tween 20) subjected to two TE-TW washes followed by 2-min centrifugation at 3000g. After removing the supernatant, beads were resuspended in 200 µl of exonuclease I mix (20 µl of 10× ExoI buffer, 10 µl of ExoI, New England Biolabs) and incubated at 37 °C for 50 min. Beads were then washed twice in TE-TW, followed by a 5 min incubation in 0.1 N NaOH at room temperature and another TE-TW wash. Second-strand synthesis was performed on beads by adding 200 µl second-strand mix (Maxima 1× reverse transcription buffer, 1 mM dNTPs, 10 mm dN-SMRT oligonucleotides, 0.125 U ml$^{-1}$ Klenow fragment) and incubating at 37 °C for 60 min. Next, beads were washed three times in TE-TW before amplification with whole transcriptome amplification PCR (1× Terra Direct PCR mix buffer, 0.25 U ml$^{-1}$ Terra polymerase, 2 mM TruSeq PCR handle primer and 2 mM SMART PCR primer) with the following conditions: 95 °C 3 min; 4 cycles of 98 °C for 20 s, 65 °C for 45 s, 72 °C for 3 min and 9 cycles of 98 °C for 20 s, 67 °C for 20 s, 72 °C for 3 min and 72 °C for 5 min. The PCR product was cleaned up by 0.6× solid-phase reversible immobilization twice and resuspended to a final volume of 10 µl. Then, 1 µl of the library was quantified on either an Agilent Bioanalyzer High Sensitivity DNA chip or Agilent TapeStation High Sensitivity D500 DNA screenTape. Then, 600 pg of the PCR product was used as input to generate Illumina sequencing libraries by tagmentation with an Illumina Nextera XT kit (catalog no. FC-131-1096). The library was amplified with TruSeq 5 and N700 series barcoded index with the following conditions: 72 °C for 3 min; 95 °C for 30 s; 12 cycles of 95 °C for 10 s, 55 °C for 30 s, 72 °C for 30 s and 72 °C for 5 min. After cleaning up, final libraries were sequenced on a NovaSeq S2 or S4 flowcells with approximately 300 million reads per array for E8.5_Embryo_1 and E9.5 embryos, approximately 200 million reads per array for E8.5_Embryo_2 and E9.0 and approximately 50 million reads per array for WT and *Tbx6* KO transversal sections.

## Slide-seq data processing and cell state annotation
The sequenced reads were processed using the Slide-seq tools pipeline (https://github.com/MacoskoLab/slideseq-tools) to generate the gene count matrix and match the bead barcode between array and sequenced reads. Most of the downstream analysis was performed in R (v.4.1.0), except the RNA velocity analysis performed in Python (v.3.8.3, v.3.9.0, v.3.10). The gene count matrix and bead spatial coordinates were processed using Seurat (v.3.0.0, v.4.0.2)[81]. Beads with more than 200 counts and less than 20% mitochondrial gene counts were retained for further analysis. The data from each stage were merged due to minimal batch effect and analyzed together. For the E8.5 replicate and E9.5 data, the top 3,000 highly variable genes were used in FindVariableFeatures and the top 40 principal components from the principal component analysis (PCA) (RunPCA from Seurat). Seurat label transfer was used to obtain cell state annotation, with the functions FindTransferAnchors and TransferData. For the second E8.5 replicate and E9.0 data, we used robust cell type decomposition (RCTD)[82] to annotate the cell state because it is more robust to lower UMI counts. Data from the E8.5 stage in our previous study were used as reference data for the Seurat label transfer function and the RCTD function[26]. Plots were generated with ggplot2 (v.3.1.0).

## Reclustering of cell types in the brain
To obtain a better understanding of the different cell types in the brain of E9.5 embryos, we performed de novo clustering from the beads corresponding to the E9.5 stage arrays, further annotating and refining the identities using the Seurat pipeline (resolution = 1), and manually annotated each of the 30 clusters using known marker genes and label transfer results.

## Differential expression analysis on brain boundaries
First, we combined marker gene spatial velocities to identify genes expressed within the brain boundaries, *Fgf8* for the mid–hindbrain boundary, *Foxg1*, *Barhl2* and *Wnt8b* for the telencephalon–diencephalon boundary, and *Barhl2* and *Pax6* for the diencephalon–midbrain boundary. Next, we calculated the distance between each bead to the boundaries and selected beads within a 300-µm distance. Genes whose expression correlated with the distance to the brain boundaries were identified using an edgeR (v.3.34.1) quasi-likelihood model (glmQLFit function)[83]. The top 40 genes ranked according to FDR were selected for heatmap visualization (Complex heatmap, v.1.99.5) in Fig. 2c.

## Identification of spatially variable genes in the developing eye
The beads associated with developing eye were identified in a semi-supervised way. First, the genes correlated with the known marker genes *Rax*, *Vax1* and *Six6* were extracted using gene co-expression analysis. These genes were used as input for the PCA and clustering pipeline in Seurat; the top 5 principal components were used. Next, we used dbscan to spatially refine the clustering results and remove a few outliers in the eye cluster. Then, we compared the eye cluster to the forebrain cluster using the FindMarkers function and searched for new marker genes specifically expressed in the developing eye regions.

## 3D reconstruction and identification of spatially variable genes in 3D
The sc3D reconstruction and associated analysis is described in detail in the Supplementary Information and can also be found at https://github.com/GuignardLab/sc3D.

## RNA velocity analysis on Slide-seq data
We adapted the scVelo (v.0.2.4) package[84] to analyze the RNA velocity at the spatial axis. Using the tutorial at https://scvelo.readthedocs.io/, exonic and intronic counts from each bead were extracted from the data and used as input. The stochastic model with default parameters was used to compute the velocity of each bead. Next, the velocity vectors were projected to the physical space. For visualization, the velocity vectors were computed according to a 50 × 50 µm grid; the 50 nearest neighbors were selected in each grid to calculate the average velocity. The length of velocity with regard to the speed of the transcriptomic changes was calculated by taking the average length of the velocity vector from the neighbors ($n = 50$) of a bead. The velocity confidence represents the coherence of the velocity direction. It was calculated by taking the sum of cosine angle between the velocity vector and its neighbors ($n = 50$). The function rank_velocity_genes was used to identify and rank genes that contribute to the vector field, which means that genes are actively transcribed and have more nascent mRNA as cells differentiate.

## Trajectory analysis in the trunk region
Beads annotated as NMPs, somites and neural tube from E8.5 and E9.5 were selected. Next, beads with a prediction score lower than 0.6 were discarded to remove cell mixtures. The remaining beads from E8.5 and E9.5 were integrated using Harmony (v.0.1.0)[85] with default parameters; then, UMAP dimensionality reduction (runUMAP) was performed based on an integrated matrix. Next, we used Monocle3 (v.1.0.0) to calculate the pseudotime from the UMAP output, according to the tutorial and using default parameters (https://cole-trapnell-lab.github.io/monocle3/docs/trajectories/). A generalized linear model with quasi-likelihood dispersion estimators from edgeR (v.3.34.1)[83] was used to find the genes that correlated with the pseudotime trajectory. Briefly, as with the tutorial[86] instructions, we used estimateDisp to estimate the gene-wise negative binomial dispersions, followed by glmQLFit or glmQLFTest to test genes that were significantly correlated with the pseudotime value, which was used as a covariate in the design

matrix. Genes with an FDR < 0.01 and a log fold change greater than 0.05 were selected and plotted into a heatmap.

### Analysis of the relationship between transcriptional dynamics and spatial distance

After trajectory analysis in the trunk region, the pairwise distance between beads within each cell state were calculated based on their spatial distance and pseudotime difference. This analysis results in the generation of a spatial and pseudotime distance matrices. Next, the spatial distance and pseudotime distance of each pair were compared and plotted.

### Identification of spatial differentially expressed genes in the neural tube

The beads assigned to have a neural tube identity in array E9.5_10 were used for this analysis. Using Slingshot::slingshot (v.2.0.0)[87], we calculated the principal curves from the beads' spatial location and ordered the cells along the principal curves as their anterior to posterior distance. The distance of each bead to the convex hull of the neural tube was computed as the dorsal to ventral distance and split into eight equally spaced bins. We used SPARK (v.1.1.1)[88] with default parameters to find spatially variable genes. We took the intersection of spatially variable genes from SPARK and the variable genes from Seurat as the input for the spatial module analysis. Dynamic time warping was applied to find genes with coherent spatial patterns that we defined as modules (in the Hox genes analysis). The Dtwclust (v.5.5.6) (https://CRAN.R-project.org/package=dtwclust) package was used for the analysis with the tsclust function ($k = 8$). The centroid of each module was calculated as their spatial pattern. Next, we used the same edgeR pipeline as the trajectory analysis to find more genes correlated with each spatial pattern. We selected genes with the same threshold and visualized them into a heatmap. The *Hox* genes were grouped as described in the previous section and the average expression of each module was smoothed using the gam::gam (v.1.2.0) function for visualization. To determine the protein classes that are enriched in the differentially and spatially variable genes, we used PANTHER (v.17.0)[89].

### Analysis of *Tbx6* mutant Slide-seq data

The data from the transversal sections of WT and *Tbx6* mutant embryos were merged and analyzed due to the low batch effect. The neural tube and somite regions were extracted based on marker gene expression, cell type labels and tissue morphology. Because of the small sample size, the extracted beads were processed using the Seurat pipeline with different parameters. The parameters used were the following: select 1,000 genes using the FindVariableFeatures function; use the first 20 principal components in the FindNeighbors function; and consider 15 neighbors in the *k*-nearest neighbor calculation, using of a resolution of 0.8 in the FindClusters function and setting n.neighbors = 20, min. dist = 0.3 in the RunUMAP function. After de novo clustering, we annotated each cluster by its marker genes and excluded cluster 0 because it mainly contained low-quality beads. Clusters 4 and 5 corresponded to the neural crest and neural plate, respectively, ruling out further analysis. We computed the raster spatial density of each cluster using MASS::kde2d (v.7.3–54) and combined them by taking the highest density for each position, which is plotted in Fig. 5b. We mapped the WT and *Tbx6* mutant data to the E9.5 trunk dataset as a reference by calculating their mutual nearest neighbor using the fast mutual nearest neighbors correction[90] method. For each bead in the *Tbx6* mutant data, we selected its ten nearest neighbors in the integrated PCA space as anchors. We computed the variance of the dimensionality reduced matrix from neighbors as a metric of projection uncertainty. The averaged UMAP coordinates of ten nearest neighbors for each bead were projected on the reference UMAP, with size representing the projection uncertainty. The FindAllMarkers function was used to find the marker genes for each cluster. We then applied the FindMarkers function

to compare two clusters (somite versus central neural tube and central neural tube versus ectopic neural tube), with the fold change of each gene.

### Statistics and reproducibility

All attempts at replicating the observations were successful (indicated below). Preselection of samples was performed, if indicated (below). No samples or data were excluded from the analysis, unless otherwise stated in the Methods. All comparisons (*Tbx6* KO) were performed with control samples from the same experiment. Sequencing and downstream processing and analysis were independent of human intervention. Blinding was not relevant because this was not an intervention study; pipelines were executed uniformly across all samples, allowing unbiased analysis. No statistical methods were used to predetermine sample sizes, but our samples are similar to those reported in previous publications[29–39].

A two-sided Wilcoxon rank-sum test was used to identify marker genes and Bonferroni correction was used for the multiple comparisons in Fig. 2c, Extended Data Figs. 6d and 10f, Supplementary Tables 5 and 6, and Supplementary Tables 8 and 9. All other tests are described in the figure legends. The FDR and *P* values used (FDR < 0.01 and log fold change greater than 0.05) are indicated in the legends.

In Extended Data Fig. 4c, elements in the box plots are as follows: middle line, median; box plot limits, upper and lower quartiles; whiskers, s.d.

The Slide-seq experiments involving WT embryos were performed on two whole E8.5, one E9.0 and 13 partial sections from three embryos at the E9.5 stage. Embryos were obtained from at least three independent isolation experiments and staged for somite count (3–5 somite pair stage for E8.5; 10–12 somite pair stage for E9.0; 15–18 somite pair stage for E9.5), which are the representative embryos shown in Extended Data Fig. 1a. The Slide-seq experiment involving WT and *Tbx6* KO embryos was performed on one (WT) and two (*Tbx6* KO) transversal sections. The *Tbx6* KO experiment was performed independently five times (in total) to verify the phenotype, which was reproduced in every embryo across all experiments. Sections were obtained from the posterior part of the trunk (the representative image of the section collected before Slide-seq is shown in Fig. 5a). Whole-mount in situ hybridization was performed once in WT E9.5 stage embryos for the indicated number (*n* = 3) and showed reproducible results (Extended Data Fig. 2b). The RNA–FISH experiments were performed from two of the independent experiments with the indicated number (*n* = 3 embryos) and showed reproducible results, with one representative image shown in Extended Data Figs. 5c, 7f, 9f and 10g, and Figs. 4e and 5e.

### Reporting summary

Further information on research design is available in the Nature Portfolio Reporting Summary linked to this article.

## Data availability

Raw and processed data can be downloaded from the Gene Expression Omnibus under accession no. GSE197353. The input object for 3D visualization for the following embryos can be downloaded: E8.5_Embryo1 (https://figshare.com/s/1c29d867bc8b90d754d2); E8.5_Embryo2 (https://figshare.com/articles/dataset/E8_5_Embryo2_h5ad/21695849/1); E9.0 (https://figshare.com/articles/dataset/E9_0_Embryo_h5ad/21695879/1). Individual Slide-seq arrays can be visualized at https://cellxgene.cziscience.com/collections/d74b6979-efba-47cd-990a-9d80ccf29055. Whole-mount in situ hybridization probe sequences and plasmids are available at http://mamep.molgen.mpg.de, with accession numbers and sequences shown in Supplementary Table 10. The FISH probe accession codes can be found in Supplementary Table 10. Complete probe sequences are the property of Molecular Instruments. See the Supplementary Note for details on tutorials and additional user information for sc3D.

## Code availability

The code used to reproduce the analyses is indexed at https://github.com/GuignardLab/sc3D for the sc3D 3D reconstruction and https://github.com/LuyiTian/Embryo_Slideseq_analysis for the E9.5 analysis. The 3D embryo can be visualized with sc3D-viewer by following the detailed instructions provided (https://github.com/GuignardLab/napari-sc3D-viewer). PuckCaller can be accessed at https://github.com/MacoskoLab/PuckCaller.

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

## Acknowledgements

We thank members of the Meissner, Chen and Macosko laboratories for critical discussion and feedback. We thank M. Scholz-Wittler and B. Hermann for sharing plasmids from the Molecular Anatomy of the Mouse Embryo Project for the in situ hybridization experiments. We thank J. Fiedler, M. Peetz, A. Landsberger and C. Franke from the transgenic animal unit of the Max Planck Institute for Molecular Genetics for their support with animal work. This work was supported by the Max Planck Society (A.M.). L.G. received funding from the Investissements d'Avenir program of the French Government managed by the French National Research Agency (ANR-16-CONV-0001) and from Excellence Initiative of Aix-Marseille University—A*MIDEX. F.C. acknowledges support from the National Institutes of Health Early Independence Award (DP5, 1DP5OD024583), the National Human Genome Research Institute (R01, R01HG010647), the Burroughs Wellcome Fund CASI award, the Searle Scholars Foundation, the Harvard Stem Cell Institute and the Merkin Institute.

## Author contributions

A.S.K., A.B., E.Z.M., F.C. and A.M. conceptualized the study. A.S.K., L.T. and R.S. performed the Slide-seq experiments with help from A.B. and E.M. L.T. performed the data analysis with help from A.S.K., H.K., A.B. and A.A.H. A.A.H. and Y.E. provided critical feedback regarding the analysis of the brain regions. L.G. conceptualized, developed and supervised the 3D reconstruction and analysis. A.S.K. and L.W. performed the transgenic animal experiments. A.S.K. performed the in situ hybridization and RNA–FISH experiments with help from M.W., G.B. and L.H. A.M., F.C. and E.Z.M. supervised the study. A.S.K., A.B. and A.M. wrote the manuscript with critical input from all authors.

## Funding

## Competing interests

E.Z.M. and F.C. are paid consultants of Atlas Biologicals. The other authors declare no competing interests.

## Additional information

**Extended data** is available for this paper at https://doi.org/10.1038/s41588-023-01435-6.

**Correspondence and requests for materials** should be addressed to Alexander Meissner.

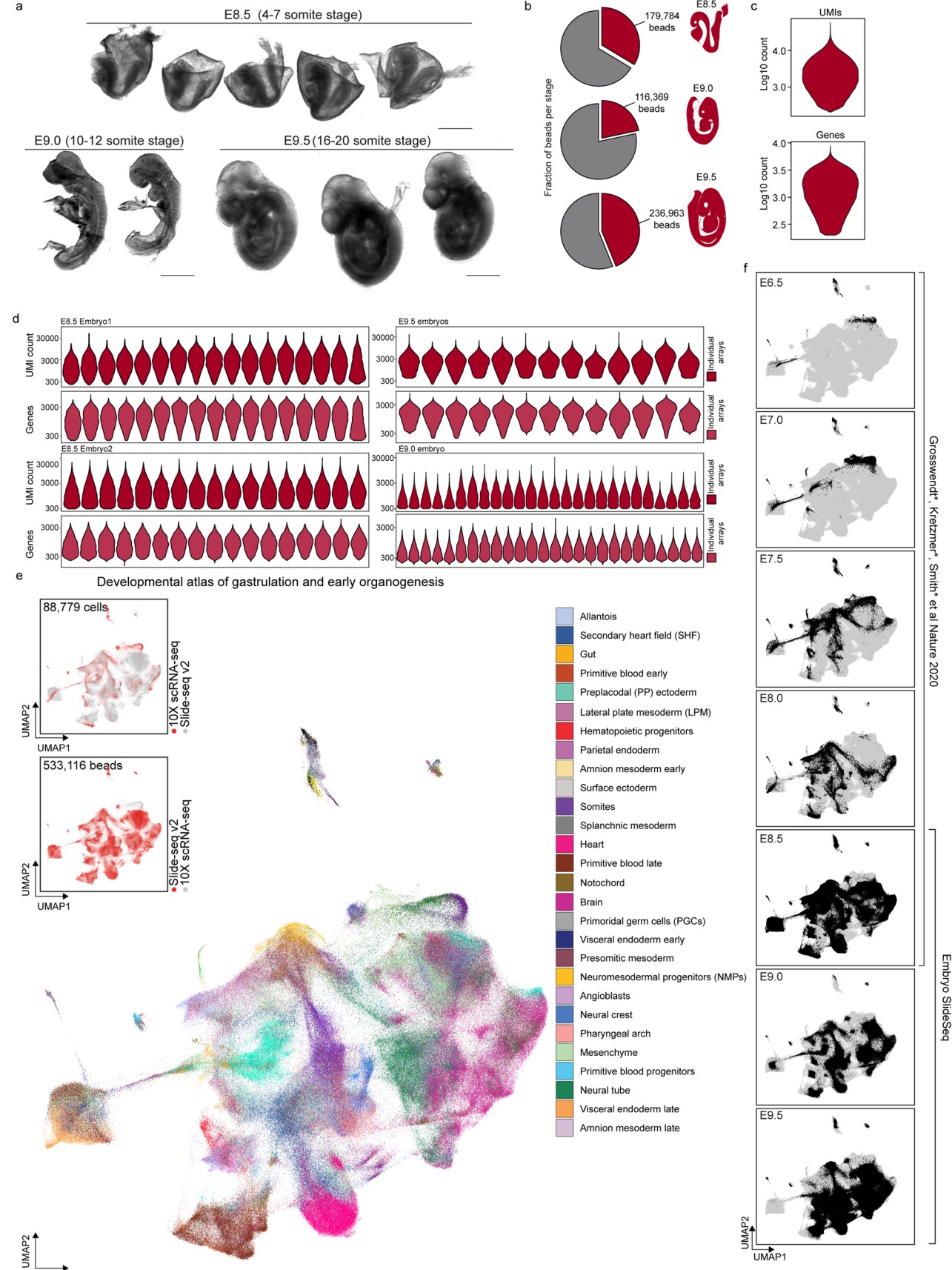

**Extended Data Fig. 1 | See next page for caption.**

**Extended Data Fig. 1 | Integration and cell-type annotation. a.** Brightfield images of representative embryos isolated from 3 independent foster mice and staged for the respective developmental stages. Scale bar, 500 μm. **b.** Distribution of beads profiled by Slide-seq from the respective stages. **c.** Violin plots showing the number of UMIs or genes recovered per bead. $Log_{10}$ values are used to represent counts. UMI, unique molecular identifier. **d.** Violin plots showing the number of UMIs and genes recovered per bead across individual arrays. $Log_{10}$ values are used to represent counts. UMI, unique molecular

identifier. **e.** Uniform Manifold Approximation and Projection (UMAP) of Slide-seq data and 10X scRNA-seq reference atlas of mouse gastrulation[26]. The color of the beads corresponds to the predicted and annotated cell state. Inset: UMAP representation of beads covered by the indicated modalities (red). Each dot represents a bead or a cell. **f.** UMAP of integrated data from stages E6.5 to E9.5. Black beads represent cells/beads from the corresponding stage. Each dot represents a bead or a cell.

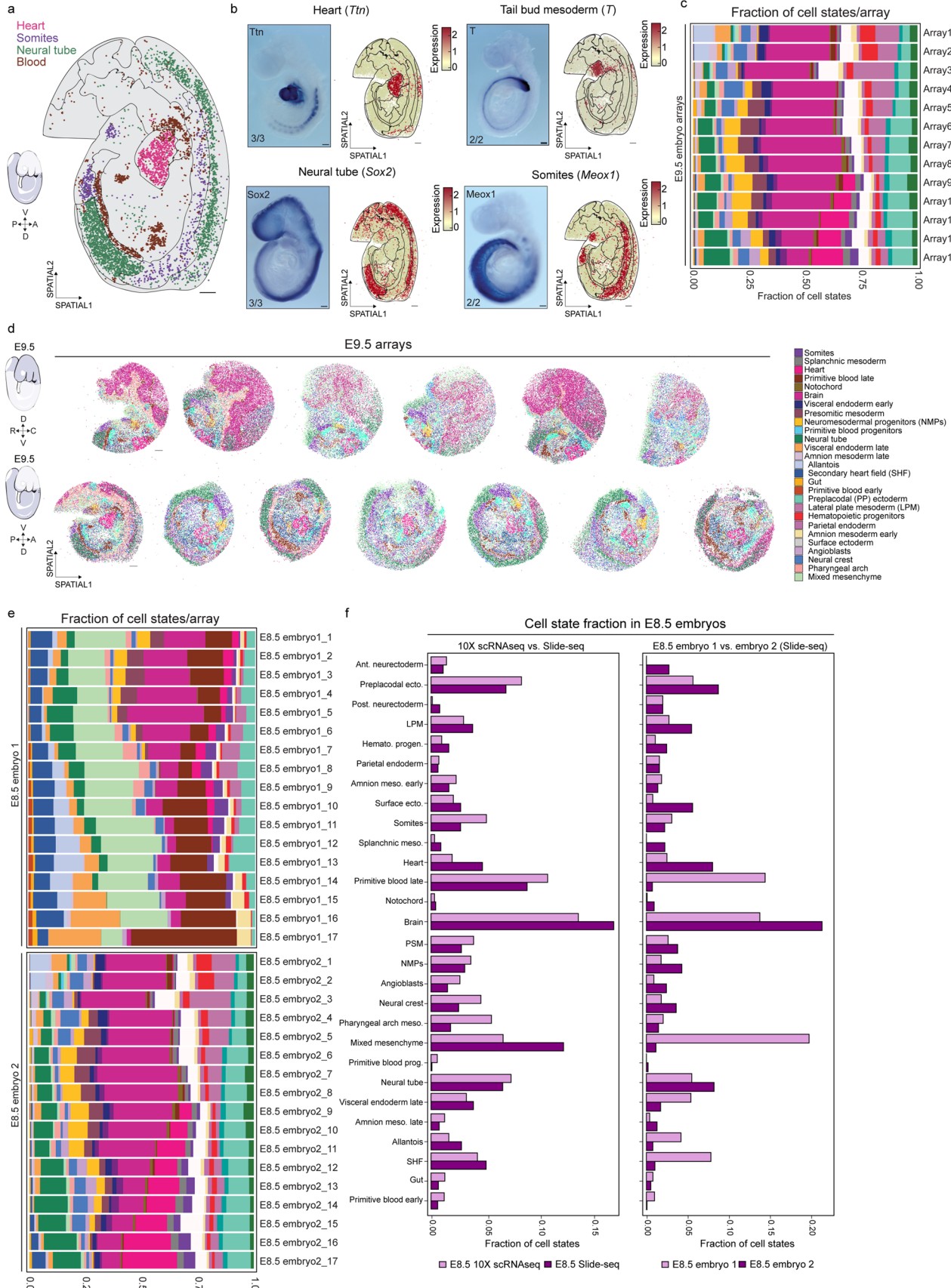

**Extended Data Fig. 2 | See next page for caption.**

**Extended Data Fig. 2 | Spatial organization of cell states. a**. Representative trunk/thoracic array (array E9.5_2) with highlighted cell states projected spatially. Each dot represents a bead. Outlines are used to emphasize morphological characteristics. A, anterior; P, posterior; D, dorsal; V, ventral. Scale bar, 200 μm **b**. Comparison of Slide-seq and conventional whole-mount *in situ* hybridization for gene expression patterns. Spatial gene expression plot showing the expression of *Ttn* (heart), *Sox2* (neural tube), *T* (tailbud mesoderm), and *Meox1* (somites). The color scale depicts normalized gene expression. n/n in the whole-mount *in situ* hybridization panel indicates the number of embryos exhibiting the pattern to the total number of embryos assayed (from one experiment). Each dot represents a bead. Scale bar, 200 μm. **c**. Cell states distribution of annotated clusters in the individual E9.5 arrays. Colors represent individual cell states, legend in panel (**d**). **d**. Spatial projection of annotated cell states in E9.5 embryo arrays. The panel depicts arrays that cover the trunk/thoracic and head regions. Each color corresponds to a distinct cell state. Each dot represents a bead. Scale bar, 200 μm. A, anterior; P, posterior; D, dorsal; V, ventral; R, rostral; C, caudal. **e**. Cell state distribution of the annotated clusters in individual arrays of the two independently profiled whole E8.5 stage embryos. Colors represent individual cell states according to the legend in panel (**d**). **f**. Cell state distribution of individual states in 10X scRNA-seq reference[26] and Slide-seq at E8.5 stage (left panel), and the comparison between the two whole E8.5 embryos profiled by Slide-seq (right panel).

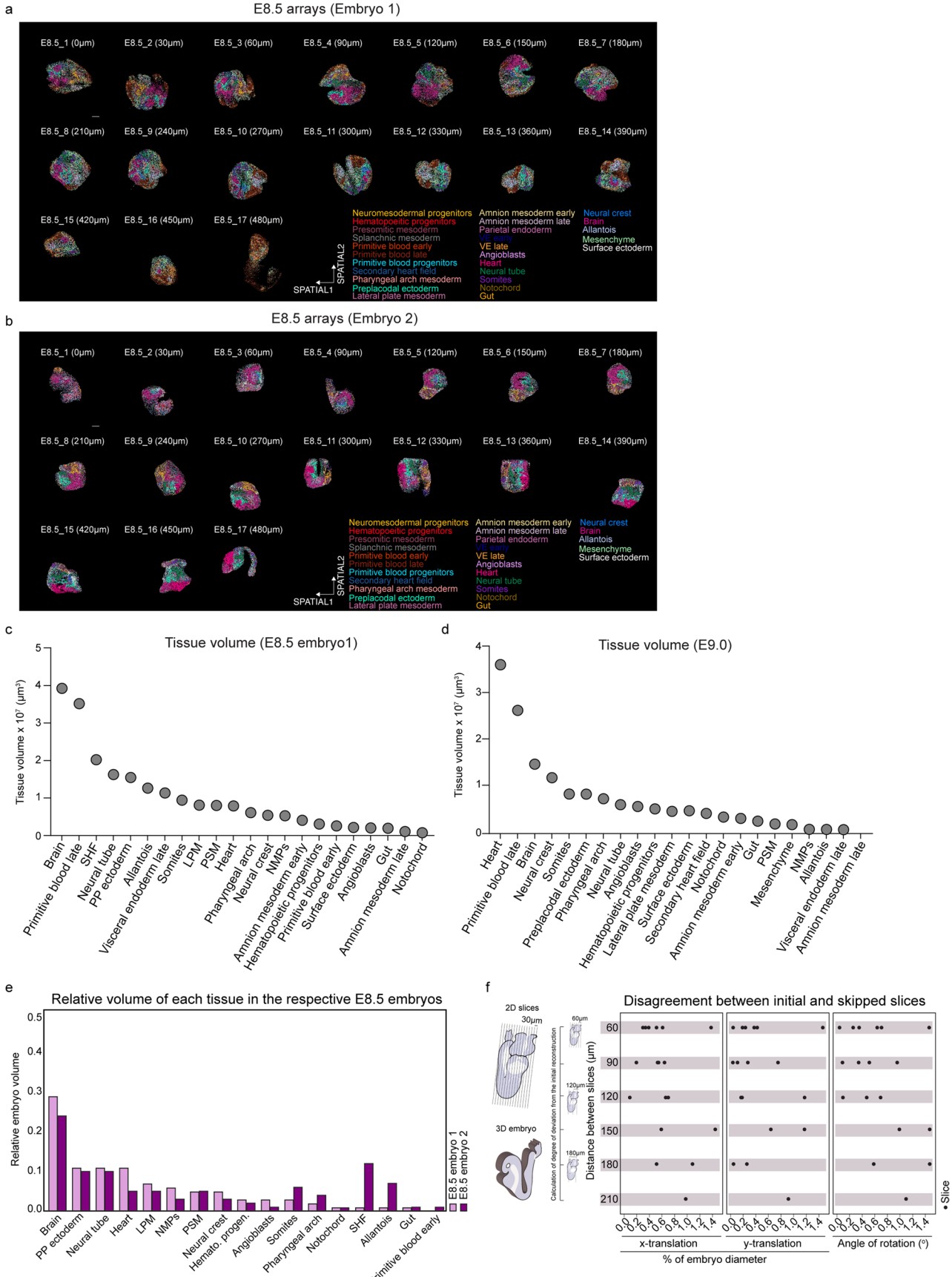

**Extended Data Fig. 3 | E8.5 and E9.0 3D embryos. a**, **b**. Spatial projection of cell states in individual arrays of the two whole E8.5 embryos. Each dot corresponds to a bead. Each color represents a cell state. Scale bar, 200 μm. **c-d**. Volumes of the indicated tissues ranked from the largest to the smallest, calculated from the E8.5 and E9.0 3D virtual embryo. Tissue volumes are listed in Supplementary Table 2. **e**. Relative volume of each tissue in the two whole E8.5 embryos. **f**. Disagreement between the initial and skipped slices when increasing the distance between individual slices.

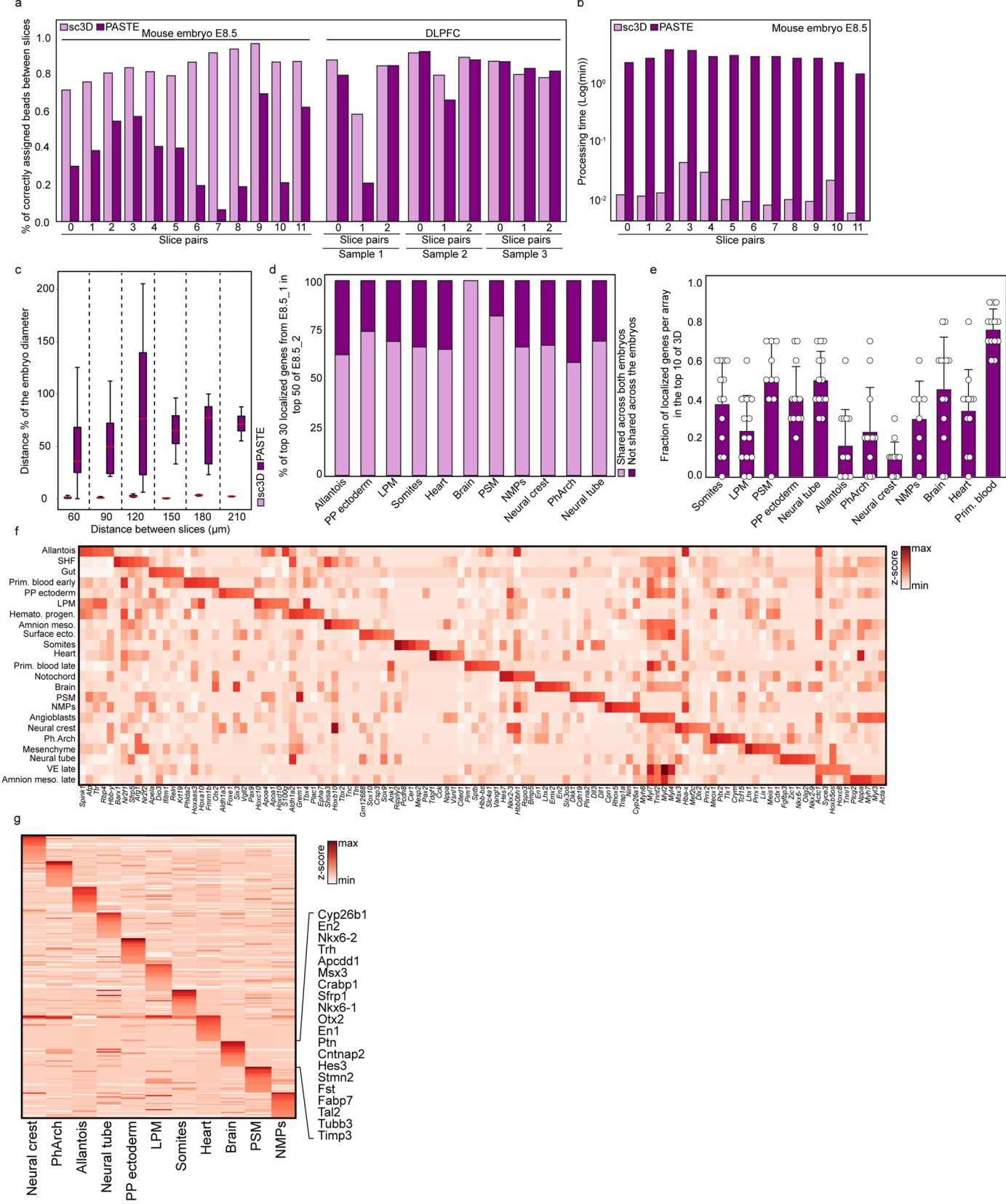

**Extended Data Fig. 4 | See next page for caption.**

**Extended Data Fig. 4 | Robustness of sc3D. a, b**. Comparison of accuracy and processing time (*sc3D* vs PASTE) across different datasets. **c**. Boxplot of the pairwise distances between the beads in the correctly registered images and those generated by the algorithm when slices are separated by 60, 90, 120, 150, 180, 210 µm ('n' slice pairs: 6,4,3,3,2,1, respectively). Boxplot elements: middle line, median; box plot limits, upper and lower quartiles; whiskers, standard deviation. **d**. Barplot of the overlap of localized genes across tissues in the two individual E8.5 whole embryos. Complete list of genes can be found in Supplementary Table 3. **e**. Barplot of the fraction of localized genes in the individual 2D arrays that were in the top 10 ranked localized genes in 3D volume of the respective tissues across all tissues in the E8.5 embryo (shown is the analysis for E8.5_Embryo_1). Error bars denote standard deviation of genes between the 2D slices and 3D volume (n = 13 genes). **f-g**. Heatmap of top localized genes (row z-score normalization) across the indicated tissues in the E8.5 embryo_1 (**f**) and E9.0 (**g**) embryo.

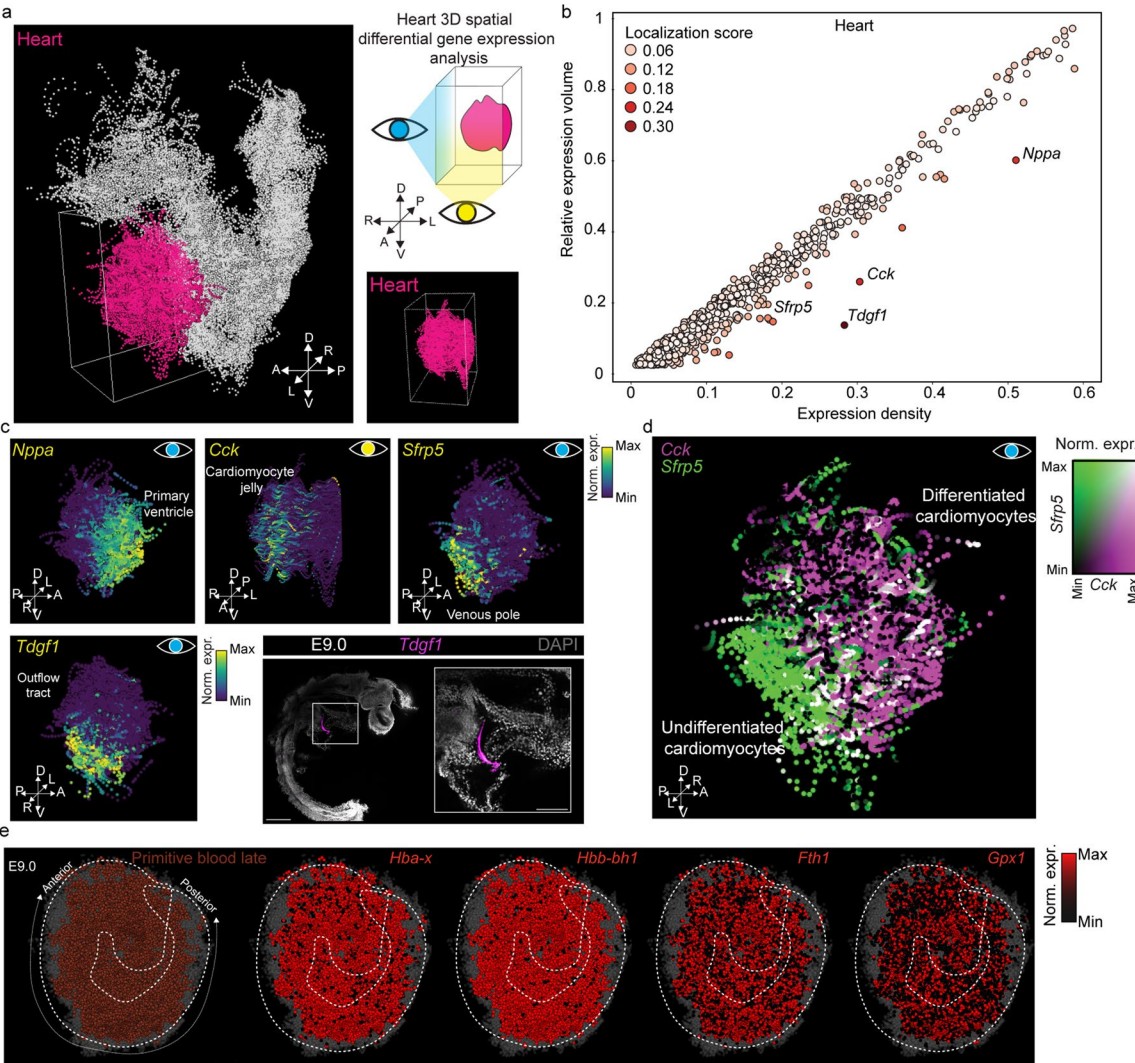

**Extended Data Fig. 5 | Localized gene expression in the developing heart.**
**a**. 3D view of E8.5 embryo highlighting the heart tissue volume (in pink). Each dot corresponds to a bead. The point of view is denoted by the eye symbol (yellow or blue). **b**. Scatter plot showing the top localized genes in the heart tissue volume. Color scale corresponds to localization score ranging from 0.06 (light red) to 0.30 (dark red). Each dot represents a gene. *x*-axis shows the density of gene expression, and *y*-axis shows the relative volume of expression within the tissue. **c**. vISH of localized genes, *Nppa, Cck, Sfrp5 and Tdgf1*. Color scale denotes the normalized

gene expression values ranging from minimum to maximum for every gene. Each dot corresponds to a bead. The point of view is denoted by the eye symbol (yellow or blue). RNA-FISH in the E9.0 embryo shows the distinct localization of *Tdgf1* in the heart. Scale bar, 100 μm. **d**. vISH showing gene co-expression for *Cck* (magenta) and *Sfrp5* (green) in the heart tissue. Each dot corresponds to a bead. Color scale for each gene ranges from black to magenta, or black to green. Beads double-positive are displayed in white. **e**. vISH for spatially ubiquitous genes in the state 'primitive blood late'. Scale bar, 200 μm. A, anterior; P, posterior; D, dorsal; V, ventral; L, left; R, right.

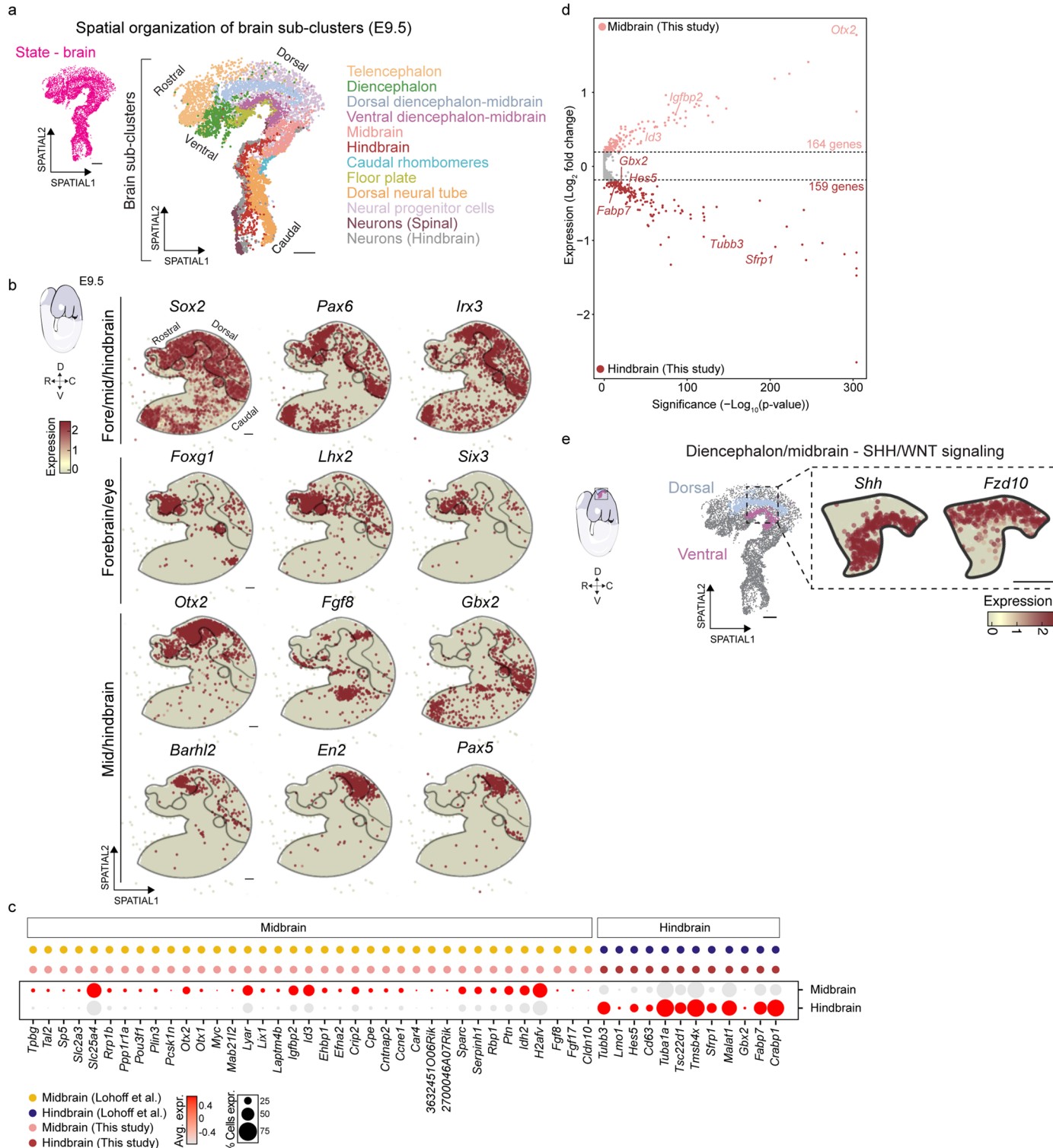

**Extended Data Fig. 6 | Spatial organization in the developing brain.**
**a**. Spatial plot of the annotated clusters/states in the brain region of an E9.5 stage embryo (array E9.5_5). Each dot corresponds to a bead. Each color represents the annotated cell state. Scale bar, 200 μm. **b**. Spatial gene expression plot depicting the indicated genes of the mentioned categories. Each dot represents a bead. The color scale depicts normalized gene expression. Scale bar, 200 μm. Array shown is E9.5_3. **c**. Dotplot showing the expression of mid and hindbrain genes (annotated[25] in the mid and hindbrain clusters in this study). Size of the dot represents the percentage of cells expressing the genes, and the color denotes

the normalized gene expression. **d**. Volcano plot of the differentially expressed genes between the annotated mid and hindbrain (this study). Each dot represents a gene, dots in grey are below the significance threshold (genes filtered by FDR < 0.01 and logFC>0.2; Two-sided Wilcoxon ran sum test). **e**. Schematic and spatial expression plot distinguishing dorsal and ventral diencephalon/midbrain regions. *Shh* marks the ventral domain, whereas *Fzd10* (Wnt receptor) marks the dorsal part. Each dot represents a bead. The color scale depicts normalized gene expression. Scale bar, 100 μm. D, dorsal; V, ventral; R, rostral; C, caudal.

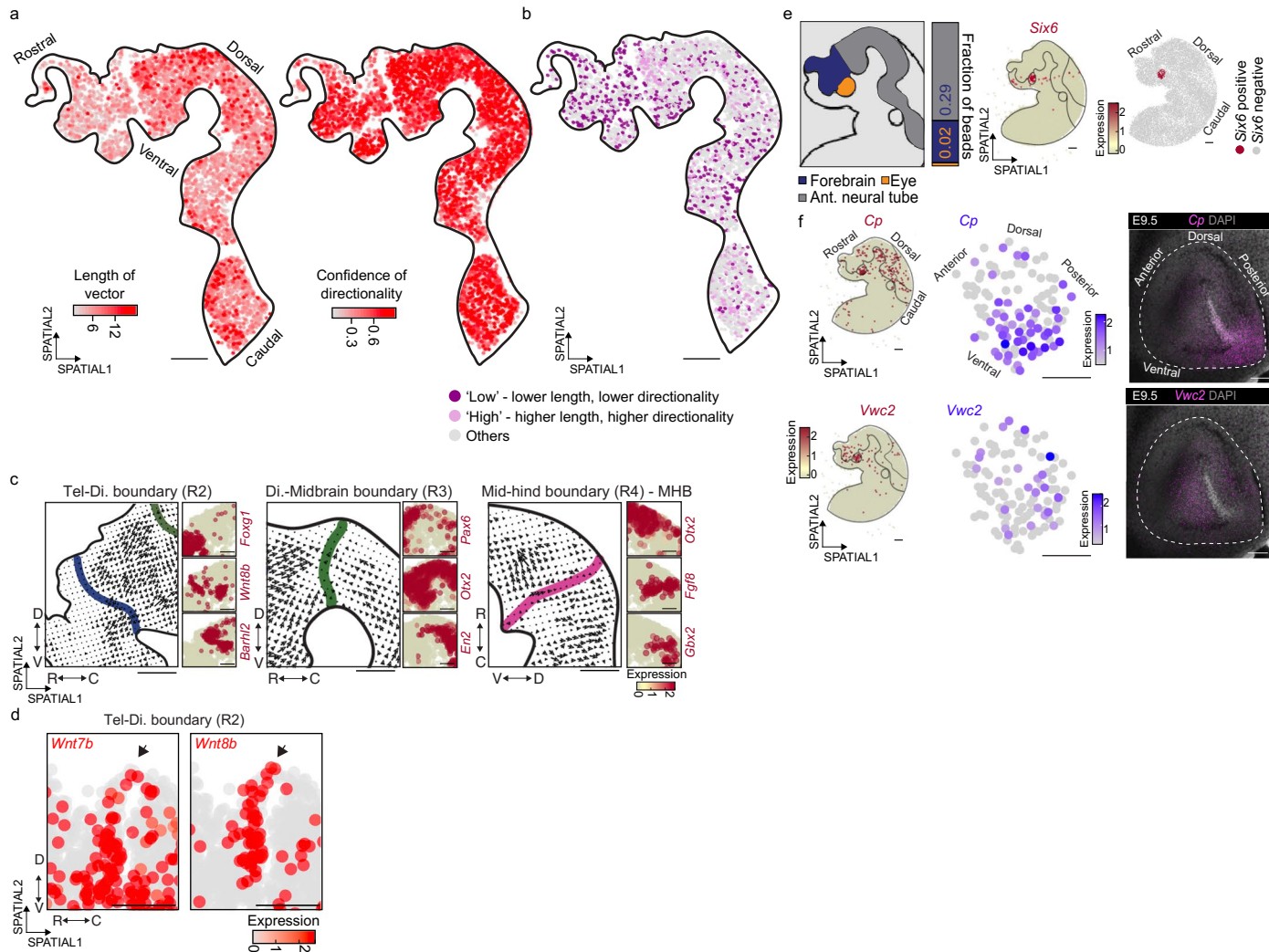

**Extended Data Fig. 7 | Patterning of the developing brain. a.** Spatial velocity length (left) and confidence of directionality (right) highlight regions with different dynamics. Scale bar, 200 μm. Array shown is E9.5_3. **b.** Combining the length and confidence measurement results in 'low' and 'high' velocity regions. Regions with 'low' velocity display lower length and lower directionality. Regions with 'high' velocity display higher length and directionality of the vector. Scale bar is 200 μm. Array shown is E9.5_3. **c.** Inset regions of the RNA velocity in the brain region with expression of markers defining the respective boundary regions (representative of 3 independent E9.5 arrays). Each dot corresponds to a bead. Color scale denotes normalized expression. Scale bar, 50 μm. **d.** Spatial plot of WNT genes at R2 (Telencephalon-diencephalon boundary – denoted arrows; representative of 3 independent E9.5 arrays). Each dot corresponds to a bead.

Color scale denotes normalized expression. Scale bar, 50 μm. **e.** Slide-seq based schematic of the developing eye. The forebrain (dark blue) and the eye (orange) are depicted with the fraction of beads corresponding to each state (bar plot). 1499/4824 beads for the forebrain/anterior neural tube; 105/4824 beads for the eye/anterior neural tube; and 105/1499 beads for the eye/forebrain. Spatial plot showing the expression of the eye-specific marker *Six6*. A, anterior; R, rostral; C, caudal; D, dorsal; V, ventral. **f.** Spatial plots showing the expression of two newly identified eye marker genes, *Cp* and *Vwc2* (left: whole E9.5_3 array; middle: subset of the 'eye'; right: RNA-FISH validations for *Cp* and *Vwc2* (magenta) and counterstained nuclei (grey) are shown for a representative embryo. n = 3 embryos/experiment, 3 independent experiments). Scalebar, 200 μm. R, rostral; C, caudal; D, dorsal; V, ventral.

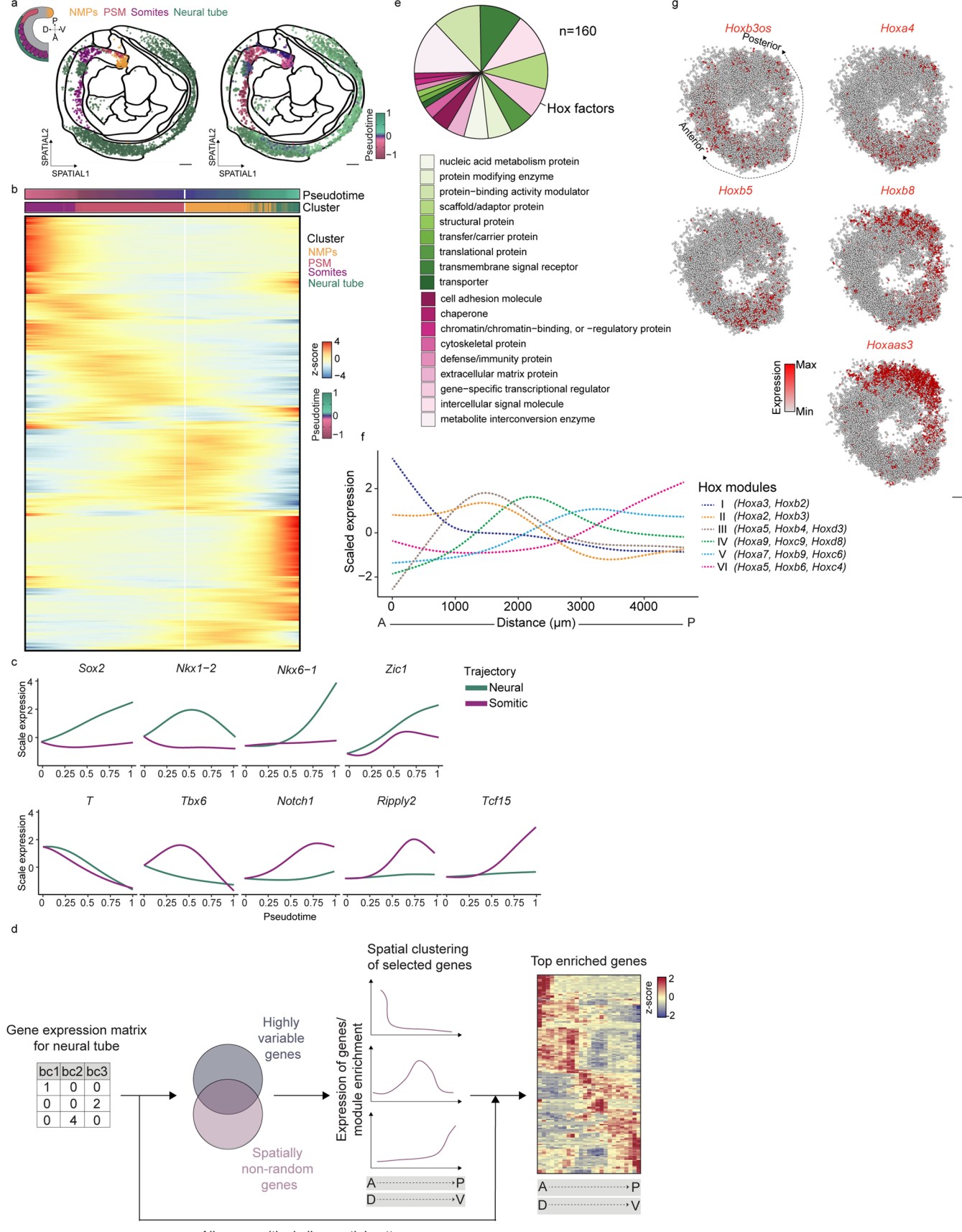

Extended Data Fig. 8 | See next page for caption.

**Extended Data Fig. 8 | Transcriptional dynamics in the trunk region.**
**a**. Spatial plot showing beads from E9.5 embryo (array E9.5_6) corresponding to the NMPs, PSM, somites and neural tube clusters (left panel) and the corresponding pseudotime values (right panel). Each dot corresponds to a bead. Cell states are highlighted in the indicated colors. Scale bar, 200 μm. **b**. Heatmap showing the expression of pseudotime determining genes along the somitic and neural trajectories. Genes used for the heatmap are listed in Supplementary Table 6. **c**. Line plot showing gene expression of selected genes along the neural

and somitic trajectories. *y*-axis represents scaled gene expression. **d**. Schematic workflow of the 'axis profiling' tool. **e**. Pie chart displaying differentially expressed genes along the anteroposterior axis divided by cellular and molecular function. Genes are listed in Supplementary Table 7. **f.** Line plot showing the spatial distribution along the anterior-posterior axis of the Hox modules' expression (I-VI). **g.** vISH for *Hox* genes in 3D virtual E9.0 stage embryo. Scale bar, 200 μm. D, dorsal; V, ventral; A, anterior; P, posterior.

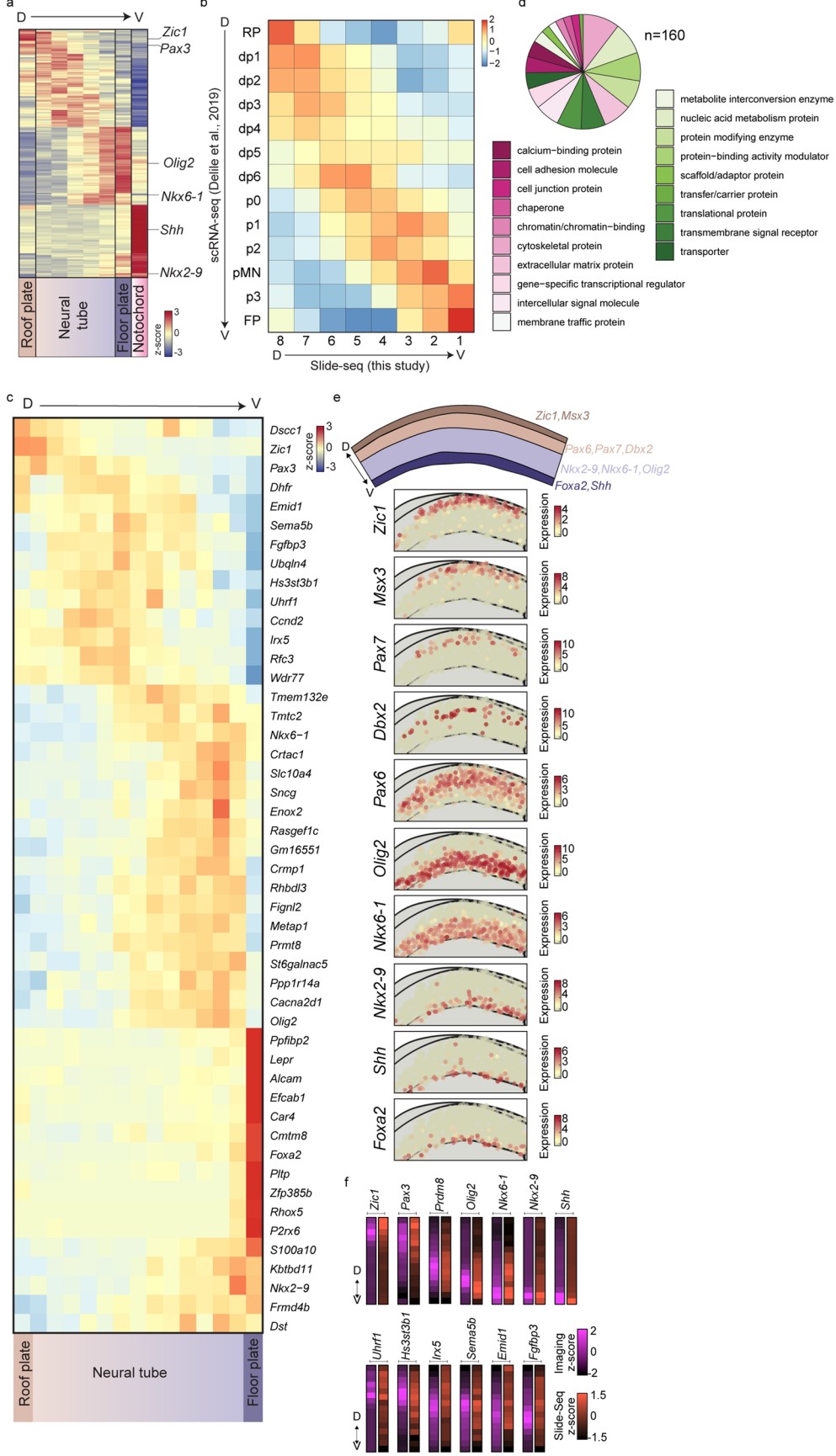

**Extended Data Fig. 9 | See next page for caption.**

**Extended Data Fig. 9 | Neural tube dorsoventral patterning. a**. Heatmap showing the top 160 genes with gene expression regionalization in one of the eight generated bins along the dorsoventral axis (genes filtered by FDR < 0.01 and logFC>0.05 then ranked by FDR). A selected set of previously known genes associated with dorsoventral patterning (black) are highlighted on the right. Specific structures along the axis are highlighted at the bottom of the heatmap. Genes are row z-score normalized and listed in Supplementary Table 7. **b**. Heatmap showing column scaled *z*-score of Pearson correlation coefficients comparing the Slide-seq dorsoventral bins and the identified clusters from a neural tube single-cell reference[26]. RP, roof plate; dp, dorsal progenitors; pMN, motor neuron progenitors; FP, floor plate. **c**. Heatmap showing an extended subset of the genes that display gene expression regionalization in one of the eight generated bins along the dorsoventral axis (genes filtered by FDR < 0.01 and logFC>0.05 then ranked by FDR). Specific structures along the axis are highlighted at the bottom of the heatmap. Genes are row z-score normalized and listed in Supplementary Table 7. **d**. Pie chart displaying differentially expressed genes along the dorsoventral axis divided by cellular and molecular function. Genes are listed in Supplementary Table 7. **e**. Schematic (top panel) and spatial gene expression plots of the known neural tube patterning genes in array E9.5_2. Each dot denotes a bead. The color scale depicts normalized gene expression. D, dorsal; V, ventral; A, anterior; P, posterior. **f**. RNA-FISH and Slide-seq quantifications for each profiled gene from Fig. 4e. Genes are bin *z*-score normalized (magenta: RNA-FISH; orange: Slide-seq). D, dorsal; V, ventral.

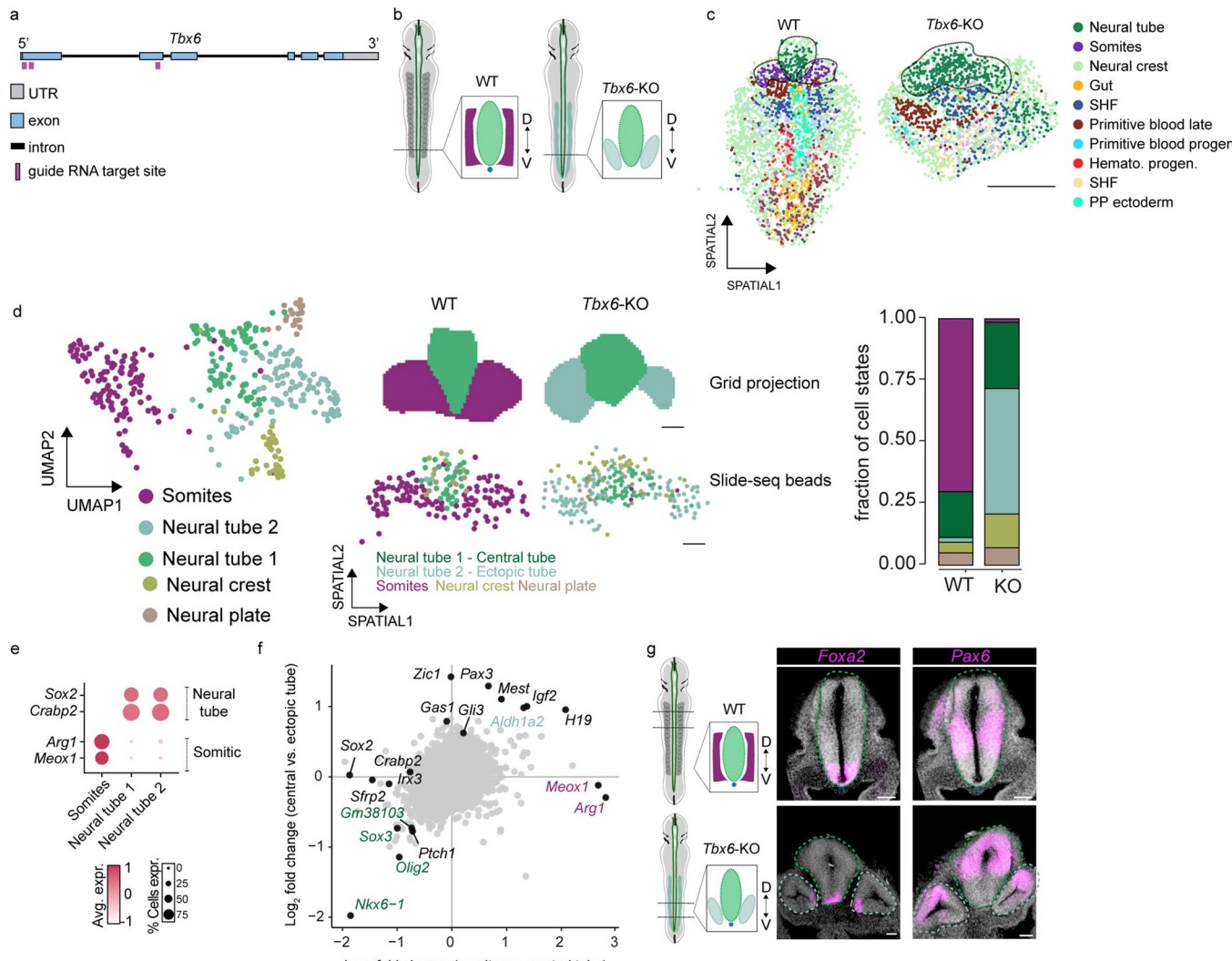

**Extended Data Fig. 10 | *Tbx6*-KO spatial profiling. a**. Schematic of *Tbx6* gene ablation strategy. Guide RNAs target the denoted exons. UTR, untranslated region. **b**. Schematic showing the plane of cryosectioning for Slide-seq experiment. **c**. Spatial plot of all cell states in the complete WT and *Tbx6*-KO trunk transversal arrays. Dotted lines denote the region that was used for further analysis. Each dot corresponds to a bead. Each color represents an individual state. Scale bar, 200 μm. **d**. UMAP, spatial and fraction plot showing the filtered and *de novo* annotated cell states. **e**. Dot plot depicting the expression of neural tube and somitic marker genes in the somites, neural tube 1 and 2 clusters.

The size of the dots represents the % of cells expressing the gene, and the color represents the cluster average normalized expression level. **f**. Scatter plot showing differentially expressed genes between somitic vs central tubes clusters and central vs ectopic tubes clusters (genes filtered by FDR < 0.01. Complete list of genes in Supplementary Table 9. **g**. RNA-FISH of *Foxa2* and *Pax6* in a transversal section of the neural tube in WT and *Tbx6*-KO embryos. The schematic represents the anteroposterior position where the transversal sections were profiled. Scale bar, 50 μm. D, dorsal; V, ventral; WT, wild type; KO, knockout.

# Reporting Summary

## Statistics

For all statistical analyses, confirm that the following items are present in the figure legend, table legend, main text, or Methods section.

| n/a | Confirmed | |
|---|---|---|
| ☐ | ☒ | The exact sample size (*n*) for each experimental group/condition, given as a discrete number and unit of measurement |
| ☐ | ☒ | A statement on whether measurements were taken from distinct samples or whether the same sample was measured repeatedly |
| ☐ | ☒ | The statistical test(s) used AND whether they are one- or two-sided *Only common tests should be described solely by name; describe more complex techniques in the Methods section.* |
| ☒ | ☐ | A description of all covariates tested |
| ☐ | ☒ | A description of any assumptions or corrections, such as tests of normality and adjustment for multiple comparisons |
| ☐ | ☒ | A full description of the statistical parameters including central tendency (e.g. means) or other basic estimates (e.g. regression coefficient) AND variation (e.g. standard deviation) or associated estimates of uncertainty (e.g. confidence intervals) |
| ☐ | ☒ | For null hypothesis testing, the test statistic (e.g. *F*, *t*, *r*) with confidence intervals, effect sizes, degrees of freedom and *P* value noted *Give P values as exact values whenever suitable.* |
| ☒ | ☐ | For Bayesian analysis, information on the choice of priors and Markov chain Monte Carlo settings |
| ☒ | ☐ | For hierarchical and complex designs, identification of the appropriate level for tests and full reporting of outcomes |
| ☐ | ☒ | Estimates of effect sizes (e.g. Cohen's *d*, Pearson's *r*), indicating how they were calculated |

*Our web collection on statistics for biologists contains articles on many of the points above.*

## Software and code

Policy information about availability of computer code

| | |
|---|---|
| Data collection | 10X Cell Ranger (v3.0), Slide-seq V2, ZEN2 (blue and black edition) |
| Data analysis | R (v3.5.1), Seurat (v3.0.0, v4.0.2), Complex Heatmap (v1.99.5), ggplot2 (v3.1.0); python (v3.8.3, v3.9.0, v3.10), scanpy (v1.4.3), velocyto python tool (v0.1.18), ImageJ (1.52p), PuckCaller, Monocle3 (1.0.0), Slingshot (2.0.0), EdgeR (3.34.1), SPARK (1.1.1), PANTHER (17.0), scVelo (0.2.4). Code used to reproduce the presented analyses is indexed: |

PuckCaller - https://github.com/MacoskoLab/PuckCaller
sc3D-3D reconstruction - https://github.com/GuignardLab/sc3D
sc3D-visualizer manual with detailed instructions - https://github.com/GuignardLab/napari-sc3D-viewer
All other analysis - https://github.com/LuyiTian/Embryo_Slideseq_analysis

For manuscripts utilizing custom algorithms or software that are central to the research but not yet described in published literature, software must be made available to editors and reviewers. We strongly encourage code deposition in a community repository (e.g. GitHub). See the Nature Portfolio guidelines for submitting code & software for further information.

## Data

Policy information about availability of data

All manuscripts must include a data availability statement. This statement should provide the following information, where applicable:
- Accession codes, unique identifiers, or web links for publicly available datasets
- A description of any restrictions on data availability
- For clinical datasets or third party data, please ensure that the statement adheres to our policy

Raw and processed data can be downloaded from GEO under accession number GSE197353. The input object for 3D visualization can be downloaded, E8.5_Embryo1 (https://figshare.com/s/1c29d867bc8b90d754d2); E8.5_Embryo2 (https://figshare.com/articles/dataset/E8_5_Embryo2_h5ad/21695849/1); E9.0 (https://figshare.com/articles/dataset/E9_0_Embryo_h5ad/21695879/1). Individual Slide-seq arrays can also be visualized at https://cellxgene.cziscience.com/collections/d74b6979-efba-47cd-990a-9d80ccf29055. Whole-mount in situ hybridization probe sequences and plasmids are available at http://mamep.molgen.mpg.de with accession codes found in Supplementary Table 10.

## Human research participants

Policy information about studies involving human research participants and Sex and Gender in Research.

| Reporting on sex and gender | n.a. |
|---|---|
| Population characteristics | n.a. |
| Recruitment | n.a. |
| Ethics oversight | n.a. |

Note that full information on the approval of the study protocol must also be provided in the manuscript.

# Field-specific reporting

Please select the one below that is the best fit for your research. If you are not sure, read the appropriate sections before making your selection.

☒ Life sciences        ☐ Behavioural & social sciences        ☐ Ecological, evolutionary & environmental sciences

For a reference copy of the document with all sections, see nature.com/documents/nr-reporting-summary-flat.pdf

# Life sciences study design

All studies must disclose on these points even when the disclosure is negative.

| Sample size | No statistical methods were used to pre-determine sample sizes, but our samples are similar to those reported in previous publications (Delile et al., Development 2019; Lohoff et al., Nat Biotechnology 2021; Peng et al., Nature 2019; Rodriques et al., Science 2019; Chen et al., Cell 2022; Stickels et al., Nat Biotechnology 2021). |
|---|---|
| Data exclusions | No data was excluded. |
| Replication | Slide-seq experiments of wild type embryos were performed on two whole E8.5, one E9.0, and 13 partial sections from 3 embryos of E9.5 stage. Embryos were obtained from at least 3 independent embryo isolation experiments and staged for somite count (3-5 somite pair stage for E8.5, 10-12 somite pair stage for E9.0, and 15-18 somite pair stage for E9.5) (representative embryos shown in Extended Data Fig.1a). Slide-seq experiment of wild type and Tbx6-KO experiment was performed on one transversal section (wild type) and two transversal sections (Tbx6-KO). Tbx6-KO experiment was performed 5 independent times to verify the phenotype, which was reproducible in every embryo across all experiments. Sections were obtained from the posterior part of the trunk (representative image of the section collected before slide-seq is shown in Fig.5a). Whole-mount in situ hybridization was performed once from wild type E9.5 stage embryos with the indicated number (n = 3) and showed reproducible results (Extended Data Fig.2b). RNA-FISH experiments were performed three times with the indicated number (n = 3 embryos), showed reproducible results, with one representative image shown in Extended Data Fig. 5c, Extended Data Fig.7f, Fig.4e, Fig.5e, Extended Data Fig.9f, and Extended Data Fig. 10i. |
| Randomization | Embryos for every experiment was staged based on morphological features corresponding to the isolation stage. |

| Blinding | Sequencing, and downstream processing and analysis, were independent of human intervention. Blinding was not relevant as it is not an intervention study, and pipelines were executed uniformly across all samples, allowing unbiased analysis. All validations and comparisons (Tbx6-KO) were performed with control samples from the same experiment. |
|---|---|

# Behavioural & social sciences study design

All studies must disclose on these points even when the disclosure is negative.

| Study description | *Briefly describe the study type including whether data are quantitative, qualitative, or mixed-methods (e.g. qualitative cross-sectional, quantitative experimental, mixed-methods case study).* |
|---|---|
| Research sample | *State the research sample (e.g. Harvard university undergraduates, villagers in rural India) and provide relevant demographic information (e.g. age, sex) and indicate whether the sample is representative. Provide a rationale for the study sample chosen. For studies involving existing datasets, please describe the dataset and source.* |
| Sampling strategy | *Describe the sampling procedure (e.g. random, snowball, stratified, convenience). Describe the statistical methods that were used to predetermine sample size OR if no sample-size calculation was performed, describe how sample sizes were chosen and provide a rationale for why these sample sizes are sufficient. For qualitative data, please indicate whether data saturation was considered, and what criteria were used to decide that no further sampling was needed.* |
| Data collection | *Provide details about the data collection procedure, including the instruments or devices used to record the data (e.g. pen and paper, computer, eye tracker, video or audio equipment) whether anyone was present besides the participant(s) and the researcher, and whether the researcher was blind to experimental condition and/or the study hypothesis during data collection.* |
| Timing | *Indicate the start and stop dates of data collection. If there is a gap between collection periods, state the dates for each sample cohort.* |
| Data exclusions | *If no data were excluded from the analyses, state so OR if data were excluded, provide the exact number of exclusions and the rationale behind them, indicating whether exclusion criteria were pre-established.* |
| Non-participation | *State how many participants dropped out/declined participation and the reason(s) given OR provide response rate OR state that no participants dropped out/declined participation.* |
| Randomization | *If participants were not allocated into experimental groups, state so OR describe how participants were allocated to groups, and if allocation was not random, describe how covariates were controlled.* |

# Ecological, evolutionary & environmental sciences study design

All studies must disclose on these points even when the disclosure is negative.

| Study description | *Briefly describe the study. For quantitative data include treatment factors and interactions, design structure (e.g. factorial, nested, hierarchical), nature and number of experimental units and replicates.* |
|---|---|
| Research sample | *Describe the research sample (e.g. a group of tagged Passer domesticus, all Stenocereus thurberi within Organ Pipe Cactus National Monument), and provide a rationale for the sample choice. When relevant, describe the organism taxa, source, sex, age range and any manipulations. State what population the sample is meant to represent when applicable. For studies involving existing datasets, describe the data and its source.* |
| Sampling strategy | *Note the sampling procedure. Describe the statistical methods that were used to predetermine sample size OR if no sample-size calculation was performed, describe how sample sizes were chosen and provide a rationale for why these sample sizes are sufficient.* |
| Data collection | *Describe the data collection procedure, including who recorded the data and how.* |
| Timing and spatial scale | *Indicate the start and stop dates of data collection, noting the frequency and periodicity of sampling and providing a rationale for these choices. If there is a gap between collection periods, state the dates for each sample cohort. Specify the spatial scale from which the data are taken* |
| Data exclusions | *If no data were excluded from the analyses, state so OR if data were excluded, describe the exclusions and the rationale behind them, indicating whether exclusion criteria were pre-established.* |
| Reproducibility | *Describe the measures taken to verify the reproducibility of experimental findings. For each experiment, note whether any attempts to repeat the experiment failed OR state that all attempts to repeat the experiment were successful.* |
| Randomization | *Describe how samples/organisms/participants were allocated into groups. If allocation was not random, describe how covariates were controlled. If this is not relevant to your study, explain why.* |
| Blinding | *Describe the extent of blinding used during data acquisition and analysis. If blinding was not possible, describe why OR explain why blinding was not relevant to your study.* |

Did the study involve field work? ☐ Yes ☐ No

## Field work, collection and transport

| | |
|---|---|
| Field conditions | *Describe the study conditions for field work, providing relevant parameters (e.g. temperature, rainfall).* |
| Location | *State the location of the sampling or experiment, providing relevant parameters (e.g. latitude and longitude, elevation, water depth).* |
| Access & import/export | *Describe the efforts you have made to access habitats and to collect and import/export your samples in a responsible manner and in compliance with local, national and international laws, noting any permits that were obtained (give the name of the issuing authority, the date of issue, and any identifying information).* |
| Disturbance | *Describe any disturbance caused by the study and how it was minimized.* |

# Reporting for specific materials, systems and methods

We require information from authors about some types of materials, experimental systems and methods used in many studies. Here, indicate whether each material, system or method listed is relevant to your study. If you are not sure if a list item applies to your research, read the appropriate section before selecting a response.

### Materials & experimental systems

| n/a | Involved in the study |
|---|---|
| ☐ | ☒ Antibodies |
| ☒ | ☐ Eukaryotic cell lines |
| ☒ | ☐ Palaeontology and archaeology |
| ☐ | ☒ Animals and other organisms |
| ☒ | ☐ Clinical data |
| ☒ | ☐ Dual use research of concern |

### Methods

| n/a | Involved in the study |
|---|---|
| ☒ | ☐ ChIP-seq |
| ☒ | ☐ Flow cytometry |
| ☒ | ☐ MRI-based neuroimaging |

## Antibodies

| | |
|---|---|
| Antibodies used | Anti-Digoxigenin-AP, Fab fragments (11093274910, Roche) |
| Validation | Antibody used in published studies (Guo et al., eLife 2022; Paulissen et al., eLife 2022; Veenvliet et al., Science 2020). |

## Eukaryotic cell lines

Policy information about cell lines and Sex and Gender in Research

| | |
|---|---|
| Cell line source(s) | *State the source of each cell line used and the sex of all primary cell lines and cells derived from human participants or vertebrate models.* |
| Authentication | *Describe the authentication procedures for each cell line used OR declare that none of the cell lines used were authenticated.* |
| Mycoplasma contamination | *Confirm that all cell lines tested negative for mycoplasma contamination OR describe the results of the testing for mycoplasma contamination OR declare that the cell lines were not tested for mycoplasma contamination.* |
| Commonly misidentified lines (See ICLAC register) | *Name any commonly misidentified cell lines used in the study and provide a rationale for their use.* |

## Palaeontology and Archaeology

| | |
|---|---|
| Specimen provenance | *Provide provenance information for specimens and describe permits that were obtained for the work (including the name of the issuing authority, the date of issue, and any identifying information). Permits should encompass collection and, where applicable, export.* |
| Specimen deposition | *Indicate where the specimens have been deposited to permit free access by other researchers.* |
| Dating methods | *If new dates are provided, describe how they were obtained (e.g. collection, storage, sample pretreatment and measurement), where* |

| Dating methods | | they were obtained (i.e. lab name), the calibration program and the protocol for quality assurance OR state that no new dates are provided. |

☐ Tick this box to confirm that the raw and calibrated dates are available in the paper or in Supplementary Information.

| Ethics oversight | | Identify the organization(s) that approved or provided guidance on the study protocol, OR state that no ethical approval or guidance was required and explain why not. |

Note that full information on the approval of the study protocol must also be provided in the manuscript.

# Animals and other research organisms

Policy information about studies involving animals; ARRIVE guidelines recommended for reporting animal research, and Sex and Gender in Research

| Laboratory animals | Animals were kept under SPF-conditions in individually ventilated cages at 22+/- 2⊠C, humidity of 55+/-10% with a 12-hour light/dark cycle (6am-6pm). IVF was performed with B6D2F1 oocyte donors (age 7-9 weeks; Envigo), and sperm isolated from B6/CAST F1 male (2 months of age; generated in-house by breeding C57BL/6J female and CAST/EiJ male). For embryo transfer experiments, pseudopregnant CD-1 female mice (Hsd:ICR; 9-12 weeks; 21-25g; Envigo) were mated with Vasectomized males (SW; >13 weeks age; Envigo). |

| Wild animals | The study did not involve wild animals. |

| Reporting on sex | Findings do not apply to one sex. Male and female embryos were used for analysis. In Slide-seq experiments, the sex was not determined prior to the experiment. |

| Field-collected samples | The study did not involve field-collected samples. |

| Ethics oversight | All procedures follow strict animal welfare guidelines as approved by the Max Planck Institute for Molecular Genetics (G0247/13-SGr1 and ZH120). |

Note that full information on the approval of the study protocol must also be provided in the manuscript.

# Clinical data

Policy information about clinical studies
All manuscripts should comply with the ICMJE guidelines for publication of clinical research and a completed CONSORT checklist must be included with all submissions.

| Clinical trial registration | Provide the trial registration number from ClinicalTrials.gov or an equivalent agency. |

| Study protocol | Note where the full trial protocol can be accessed OR if not available, explain why. |

| Data collection | Describe the settings and locales of data collection, noting the time periods of recruitment and data collection. |

| Outcomes | Describe how you pre-defined primary and secondary outcome measures and how you assessed these measures. |

# Dual use research of concern

Policy information about dual use research of concern

## Hazards

Could the accidental, deliberate or reckless misuse of agents or technologies generated in the work, or the application of information presented in the manuscript, pose a threat to:

No | Yes
☐ | ☐ Public health
☐ | ☐ National security
☐ | ☐ Crops and/or livestock
☐ | ☐ Ecosystems
☐ | ☐ Any other significant area

## Experiments of concern

Does the work involve any of these experiments of concern:

No Yes

☐ ☐ Demonstrate how to render a vaccine ineffective

☐ ☐ Confer resistance to therapeutically useful antibiotics or antiviral agents

☐ ☐ Enhance the virulence of a pathogen or render a nonpathogen virulent

☐ ☐ Increase transmissibility of a pathogen

☐ ☐ Alter the host range of a pathogen

☐ ☐ Enable evasion of diagnostic/detection modalities

☐ ☐ Enable the weaponization of a biological agent or toxin

☐ ☐ Any other potentially harmful combination of experiments and agents

# ChIP-seq

## Data deposition

☐ Confirm that both raw and final processed data have been deposited in a public database such as GEO.

☐ Confirm that you have deposited or provided access to graph files (e.g. BED files) for the called peaks.

| | |
|---|---|
| Data access links<br>*May remain private before publication.* | *For "Initial submission" or "Revised version" documents, provide reviewer access links. For your "Final submission" document, provide a link to the deposited data.* |
| Files in database submission | *Provide a list of all files available in the database submission.* |
| Genome browser session<br>(e.g. UCSC) | *Provide a link to an anonymized genome browser session for "Initial submission" and "Revised version" documents only, to enable peer review. Write "no longer applicable" for "Final submission" documents.* |

## Methodology

| | |
|---|---|
| Replicates | *Describe the experimental replicates, specifying number, type and replicate agreement.* |
| Sequencing depth | *Describe the sequencing depth for each experiment, providing the total number of reads, uniquely mapped reads, length of reads and whether they were paired- or single-end.* |
| Antibodies | *Describe the antibodies used for the ChIP-seq experiments; as applicable, provide supplier name, catalog number, clone name, and lot number.* |
| Peak calling parameters | *Specify the command line program and parameters used for read mapping and peak calling, including the ChIP, control and index files used.* |
| Data quality | *Describe the methods used to ensure data quality in full detail, including how many peaks are at FDR 5% and above 5-fold enrichment.* |
| Software | *Describe the software used to collect and analyze the ChIP-seq data. For custom code that has been deposited into a community repository, provide accession details.* |

# Flow Cytometry

## Plots

Confirm that:

☐ The axis labels state the marker and fluorochrome used (e.g. CD4-FITC).

☐ The axis scales are clearly visible. Include numbers along axes only for bottom left plot of group (a 'group' is an analysis of identical markers).

☐ All plots are contour plots with outliers or pseudocolor plots.

☐ A numerical value for number of cells or percentage (with statistics) is provided.

## Methodology

| | |
|---|---|
| Sample preparation | *Describe the sample preparation, detailing the biological source of the cells and any tissue processing steps used.* |
| Instrument | *Identify the instrument used for data collection, specifying make and model number.* |

| Software | *Describe the software used to collect and analyze the flow cytometry data. For custom code that has been deposited into a community repository, provide accession details.* |
|---|---|
| Cell population abundance | *Describe the abundance of the relevant cell populations within post-sort fractions, providing details on the purity of the samples and how it was determined.* |
| Gating strategy | *Describe the gating strategy used for all relevant experiments, specifying the preliminary FSC/SSC gates of the starting cell population, indicating where boundaries between "positive" and "negative" staining cell populations are defined.* |

☐ Tick this box to confirm that a figure exemplifying the gating strategy is provided in the Supplementary Information.

# Magnetic resonance imaging

## Experimental design

| Design type | *Indicate task or resting state; event-related or block design.* |
|---|---|
| Design specifications | *Specify the number of blocks, trials or experimental units per session and/or subject, and specify the length of each trial or block (if trials are blocked) and interval between trials.* |
| Behavioral performance measures | *State number and/or type of variables recorded (e.g. correct button press, response time) and what statistics were used to establish that the subjects were performing the task as expected (e.g. mean, range, and/or standard deviation across subjects).* |

## Acquisition

| Imaging type(s) | *Specify: functional, structural, diffusion, perfusion.* |
|---|---|
| Field strength | *Specify in Tesla* |
| Sequence & imaging parameters | *Specify the pulse sequence type (gradient echo, spin echo, etc.), imaging type (EPI, spiral, etc.), field of view, matrix size, slice thickness, orientation and TE/TR/flip angle.* |
| Area of acquisition | *State whether a whole brain scan was used OR define the area of acquisition, describing how the region was determined.* |

Diffusion MRI     ☐ Used     ☐ Not used

## Preprocessing

| Preprocessing software | *Provide detail on software version and revision number and on specific parameters (model/functions, brain extraction, segmentation, smoothing kernel size, etc.).* |
|---|---|
| Normalization | *If data were normalized/standardized, describe the approach(es): specify linear or non-linear and define image types used for transformation OR indicate that data were not normalized and explain rationale for lack of normalization.* |
| Normalization template | *Describe the template used for normalization/transformation, specifying subject space or group standardized space (e.g. original Talairach, MNI305, ICBM152) OR indicate that the data were not normalized.* |
| Noise and artifact removal | *Describe your procedure(s) for artifact and structured noise removal, specifying motion parameters, tissue signals and physiological signals (heart rate, respiration).* |
| Volume censoring | *Define your software and/or method and criteria for volume censoring, and state the extent of such censoring.* |

## Statistical modeling & inference

| Model type and settings | *Specify type (mass univariate, multivariate, RSA, predictive, etc.) and describe essential details of the model at the first and second levels (e.g. fixed, random or mixed effects; drift or auto-correlation).* |
|---|---|
| Effect(s) tested | *Define precise effect in terms of the task or stimulus conditions instead of psychological concepts and indicate whether ANOVA or factorial designs were used.* |

Specify type of analysis:     ☐ Whole brain     ☐ ROI-based     ☐ Both

| Statistic type for inference<br>(See Eklund et al. 2016) | *Specify voxel-wise or cluster-wise and report all relevant parameters for cluster-wise methods.* |
|---|---|
| Correction | *Describe the type of correction and how it is obtained for multiple comparisons (e.g. FWE, FDR, permutation or Monte Carlo).* |

## Models & analysis

nature portfolio | reporting summary

| n/a | Involved in the study |
|-----|------------------------|
| ☐ ☐ | Functional and/or effective connectivity |
| ☐ ☐ | Graph analysis |
| ☐ ☐ | Multivariate modeling or predictive analysis |

**Functional and/or effective connectivity**

*Report the measures of dependence used and the model details (e.g. Pearson correlation, partial correlation, mutual information).*

**Graph analysis**

*Report the dependent variable and connectivity measure, specifying weighted graph or binarized graph, subject- or group-level, and the global and/or node summaries used (e.g. clustering coefficient, efficiency, etc.).*

**Multivariate modeling and predictive analysis**

*Specify independent variables, features extraction and dimension reduction, model, training and evaluation metrics.*

