## [Peer Review File · Nature Genetics]

Peer Review Information

Manuscript Title: Spatiotemporal transcriptomic maps of whole mouse embryos at the onset of organogenesis

Corresponding author name(s): Dr Evan (Z.) Macosko Dr Alexander Meissner Dr Fei Chen Dr Léo Guignard

Editorial Notes:

Transferred manuscripts (no peer review at Nature Genetics) This manuscript has been previously reviewed at another journal that is not operating a transparent peer review scheme. The manuscript was considered suitable for publication without further review at Nature Genetics

Reviewer Comments & Decisions:

Decision Letter, initial version:

9th Mar 2023

Dear Alex,

Thank you for submitting your revised manuscript "Spatial transcriptomic maps of whole mouse embryos" (NG-A61942-T). My colleagues and I find that the paper has improved in revision, and therefore we'll be happy in principle to publish it in Nature Genetics, pending minor revisions to comply with our editorial and formatting guidelines.

Thank you again for your interest in Nature Genetics. Please do not hesitate to contact me if you have any questions.

Congratulations!

Sincerely,

Tiago

Tiago Faial, PhD
Chief Editor

Nature Genetics
<https://orcid.org/0000-0003-0864-1200>

Final Decision Letter:

25th May 2023

Dear Alex,

I am delighted to say that your manuscript "Spatiotemporal transcriptomic maps of whole mouse embryos at the onset of organogenesis" has been accepted for publication in an upcoming issue of Nature Genetics.

Your paper will be published online after we receive your corrections and will appear in print in the next available issue. You can find out your date of online publication by contacting the Nature Press Office (press@nature.com) after sending your e-proof corrections. Now is the time to inform your Public Relations or Press Office about your paper, as they might be interested in promoting its publication. This will allow them time to prepare an accurate and satisfactory press release. Include your manuscript tracking number (NG-A61942R) and the name of the journal, which they will need when they contact our Press Office.

Please note that Nature Genetics is a Transformative Journal (TJ). Authors may publish their research with us through the traditional subscription access route or make their paper immediately open access through payment of an article-processing charge (APC). Authors will not be required to

make a final decision about access to their article until it has been accepted. [Find out more about Transformative Journals](https://www.springernature.com/gp/open-research/transformative-journals)

Authors may need to take specific actions to achieve [compliance with funder and institutional open access mandates](https://www.springernature.com/gp/open-research/funding/policy-compliance-faqs). If your research is supported by a funder that requires immediate open access (e.g. according to [Plan S principles](https://www.springernature.com/gp/open-research/plan-s-compliance)) then you should select the gold OA route, and we will direct you to the compliant route where possible. For authors selecting the subscription publication route, the journal's standard licensing terms will need to be accepted, including [self-archiving-and-license-to-publish](https://www.nature.com/nature-portfolio/editorial-policies/self-archiving-and-license-to-publish). Those licensing terms will supersede any other terms that the author or any third party may assert apply to any version of the manuscript.

Please note that Nature Portfolio offers an immediate open access option only for papers that were first submitted after 1 January, 2021.

Sincerely,

Tiago

Tiago Faial, PhD
Chief Editor
Nature Genetics
<https://orcid.org/0000-0003-0864-1200>